# Structure-based characterization of novel TRPV5 inhibitors

Taylor ET Hughes[1†], John Smith Del Rosario[2†], Abhijeet Kapoor[3†],
Aysenur Torun Yazici[2], Yevgen Yudin[2], Edwin C Fluck III[1], Marta Filizola[3]*,
Tibor Rohacs[2]*, Vera Y Moiseenkova-Bell[1]*

[1]Department of Systems Pharmacology and Translational Therapeutics, Perelman School of Medicine, University of Pennsylvania, Philadelphia, United States; [2]Department of Pharmacology, Physiology and Neuroscience, New Jersey Medical School, Rutgers University, Newark, United States; [3]Department of Pharmacological Sciences, Icahn School of Medicine at Mount Sinai, New York, United States

**Abstract** Transient receptor potential vanilloid 5 (TRPV5) is a highly calcium selective ion channel that acts as the rate-limiting step of calcium reabsorption in the kidney. The lack of potent, specific modulators of TRPV5 has limited the ability to probe the contribution of TRPV5 in disease phenotypes such as hypercalcemia and nephrolithiasis. Here, we performed structure-based virtual screening (SBVS) at a previously identified TRPV5 inhibitor binding site coupled with electrophysiology screening and identified three novel inhibitors of TRPV5, one of which exhibits high affinity, and specificity for TRPV5 over other TRP channels, including its close homologue TRPV6. Cryo-electron microscopy of TRPV5 in the presence of the specific inhibitor and its parent compound revealed novel binding sites for this channel. Structural and functional analysis have allowed us to suggest a mechanism of action for the selective inhibition of TRPV5 and lay the groundwork for rational design of new classes of TRPV5 modulators.
DOI: https://doi.org/10.7554/eLife.49572.001

*For correspondence:
marta.filizola@mssm.edu (MF);
rohacsti@njms.rutgers.edu (TR);
vmb@pennmedicine.upenn.edu
(VYM-B)

†These authors contributed
equally to this work

**Competing interests:** The
authors declare that no
competing interests exist.

**Reviewing editor:** Baron
Chanda, University of Wisconsin–
Madison, United States

## Introduction

TRPV5 is a calcium selective ion channel that is responsible for the fine tuning of calcium reabsorption in the kidney and has been shown to be a critical component of systemic calcium homeostasis (*Peng et al., 2000*; *Hoenderop et al., 2003*; *de Groot et al., 2008*; *Na and Peng, 2014*). Due to its role in calcium handling, potent and specific chemical modulators of TRPV5 have the potential to aid in treatments of calcium homeostasis disorders such as nephrolithiasis and hypercalcemia (*Nie et al., 2016*). By uncovering specific modulators of TRPV5, it would be possible to identify the contribution of TRPV5 to these disease phenotypes and provide a foundation of targeted treatments. Additionally, chemical probes of TRPV5 could aid in identifying its roles in cell types where it is expressed in low abundance or in cell types that express both TRPV5 and its close homologue TRPV6. While there are some synthetic modulators of TRPV5 including econazole (*Nilius et al., 2001*; *Hughes et al., 2018a*), TH-1177 (*Landowski et al., 2011*; *Peng et al., 2018*), and select cannabinoids (*Janssens et al., 2018*), all have reported potencies in the mid-micromolar range and currently none have robust selectivity for TRPV5 over TRPV6 and the other TRPV subfamily channels (*Nilius et al., 2001*; *Landowski et al., 2011*; *Peng et al., 2018*; *Janssens et al., 2018*).

Mechanisms of endogenous regulation of TRPV5 have been well documented using both structural and functional methods (*Peng et al., 2000*; *de Groot et al., 2008*; *Hughes et al., 2018a*). In the cell, binding of PI(4,5)P$_2$ is critical for channel opening while rapid desensitization occurs via binding of calmodulin (CaM) to the base of the pore (*Na and Peng, 2014*; *Hughes et al., 2018a*;

*van der Wijst et al., 2017*; *Lee et al., 2005*). Thus, activators that bind directly to TRPV5 could act by allosterically enhancing conductance during PI(4,5)P$_2$ activation, mimicking PI(4,5)P$_2$ activation or decreasing CaM-induced desensitization. On the other hand, inhibitors that bind to TRPV5 are expected to act by directly blocking PI(4,5)P$_2$ binding, allosterically limiting the conformational changes induced by PI(4,5)P$_2$ binding, directly blocking ion flow by binding in the pore or by increasing the desensitization of TRPV5 by CaM. Due to the conserved nature of PI(4,5)P$_2$ modulation within the TRP channel family, allosteric modulators are more likely to be both potent and specific for TRPV5 (*Rohács et al., 2005*; *Thyagarajan et al., 2008*; *Rohacs, 2014*).

With the availability of large compound libraries in ready-to-dock format and recent advancements in docking algorithms, structure-based virtual screening (SBVS) technology is contributing more significantly to drug discovery efforts, especially when high-resolution protein structures are available (*Lyu et al., 2019*; *Basith et al., 2018*). Here, we screened the previously identified econazole-binding pocket of TRPV5 (*Hughes et al., 2018b*), which was reported in the same location as the vanilloid binding pocket in TRPV1 (*Hughes et al., 2018b*; *Gao et al., 2016*). This allosteric binding site has been reported to bind modulators such as small molecules and lipids in several TRPV subfamily channels (*Hughes et al., 2018a*; *Hughes et al., 2018b*; *Gao et al., 2016*). Through this screen we identified three novel synthetic inhibitors that to our best knowledge have not been previously characterized as modulators for any protein.

## Results

### In silico compound screening

Currently, several structures of TRPV5 are available with a variety of modulators bound (*Hughes et al., 2018a*; *Hughes et al., 2018b*). The econazole-bound TRPV5 structure revealed that the small-molecule inhibitor of TRPV5, econazole, bound in the 'vanilloid' pocket of TRPV5, which is a well-documented site of modulator binding in other TRPV channels (*Hughes et al., 2018b*; *Gao et al., 2016*). This pocket, which was resolved to a higher resolution (~3.0 Å local resolution) in the lipid-bound TRPV5 (*Hughes et al., 2018a*), was used for SBVS. Specifically, this binding pocket is located at the interface of two adjacent monomers and is composed of the S3, S4 and S4-S5 linker of one monomer and the S5 and S6 helices of the second monomer (*Figure 1—figure supplement 1*). The resolution of this pocket in the lipid-bound TRPV5 structure allowed for unambiguous backbone placement and confident side chain placement. Using the Schrodinger suite 2018–1,~12 million compounds from the ZINC15 'Drugs Now' library, were docked into this pocket. The top 100 compounds as ranked by their docking score were grouped into 65 clusters of unique chemical scaffolds based on their chemical similarity assessed by Tanimoto similarity scores of ECFP4 binary fingerprints (*Figure 1—source data 1*). Econazole was not part of the 'in stock' library and was therefore excluded from the original screen. A follow up screen in the same pocket that included econazole revealed a less favorable docking score (−5.9) compared to the top 100 hits (<−9.0). Additionally, none of these top 100 compounds exhibited significant similarity (defined by a Tanimoto similarity score larger than 0.4) with econazole or other known TRPV5 inhibitors. Of the 65 unique chemical scaffolds identified as cluster representatives (centroid of clusters), 43 compounds were purchased based on availability and price (*Figure 1—source data 2*).

### Functional validation of compound hits

These 43 compounds were screened in HEK293 cells expressing rabbit TRPV5 using whole cell patch clamp experiments to measure modulation of monovalent currents through TRPV5, as describe earlier (*Hughes et al., 2018b*). While TRPV5 is a Ca$^{2+}$ selective ion channel, it conducts monovalent currents in the absence of Ca$^{2+}$ and Mg$^{2+}$. Ca$^{2+}$ currents are not only smaller than monovalent currents, but they also undergo Ca$^{2+}$-induced inactivation, therefore monovalent currents are routinely used for assessing channel function (*Hoenderop et al., 2001*; *Velisetty et al., 2016*). Each compound was tested at 10 µM or 3 µM and 41 of them showed no clear inhibition or potentiation of TRPV5 currents (*Figure 1—source data 2* and *Figure 1—figure supplement 2*). Some compounds may have shown no effect due to an inability to cross the membrane rather than a lack of activity on TRPV5.

Two of the 43 screened compounds measurably inhibited TRPV5-mediated currents in our system. One of the hit compounds, ZINC9155420 showed robust (~80%) inhibition of rabbit TRPV5 at 10 μM, and did not appear to have any selectivity for TRPV5 over the closely related TRPV6 channel (*Figure 1A–C*). In HEK293 cells, the $IC_{50}$ of ZINC9155420 for rbTRPV5 inhibition was 2.91 ± 0.56 μM, which is of comparable potency to econazole (*Nilius et al., 2001*; *Hughes et al., 2018b*). This

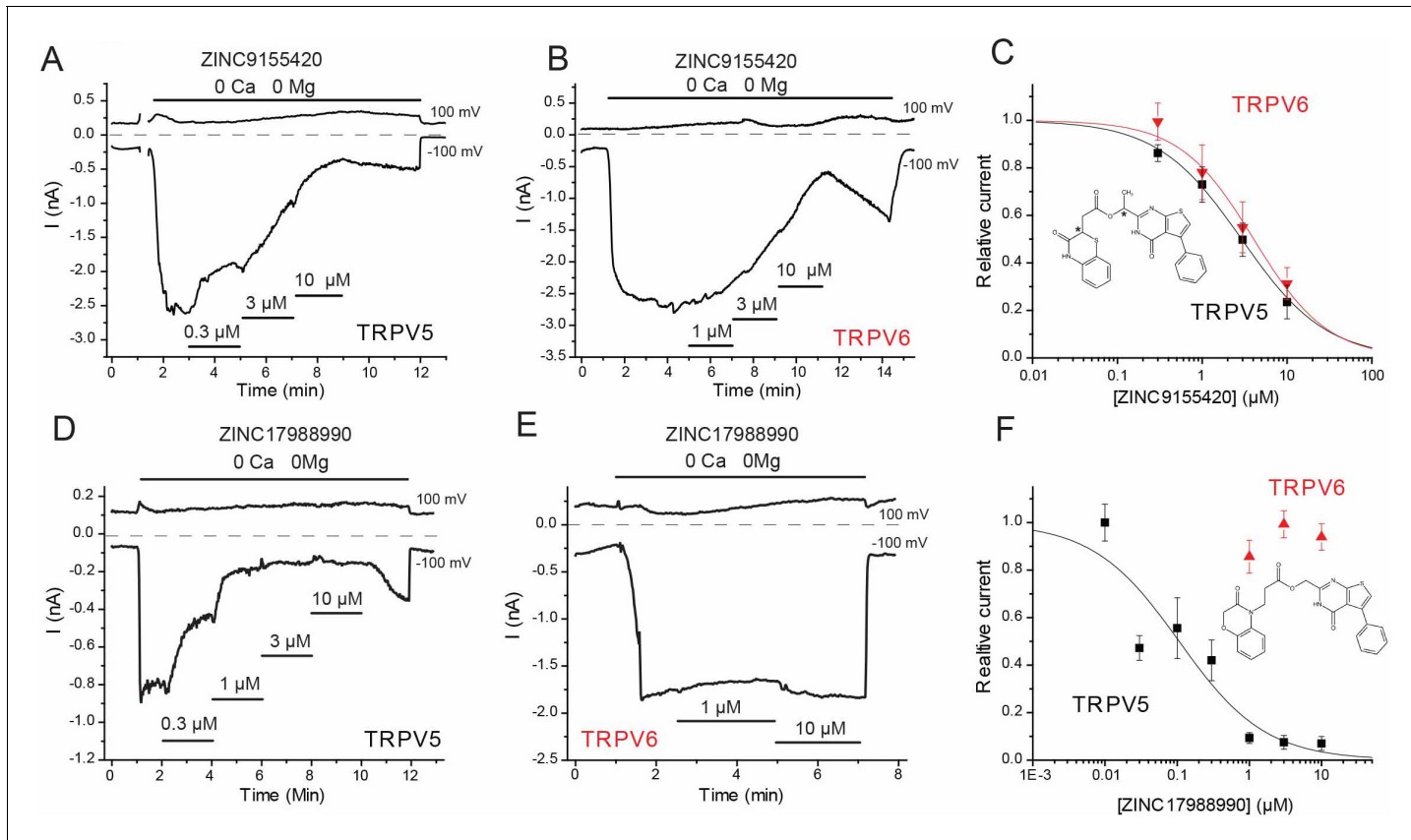

**Figure 1.** Effects of novel inhibitors on TRPV5 and TRPV6 activity. HEK293 cells were transfected with rabbit TRPV5 or human TRPV6. Whole cell patch clamp experiments were performed as described in the methods section; monovalent currents were initiated using a divalent cation free solution. Currents are shown at 100 and −100 mV; zero currents are indicated by the dashed lines. (**A–B**) Representative traces for the effects of ZINC9155420 on (**A**) TRPV5 and (**B**) TRPV6 currents, the applications of various concentrations of ZINC9155420 are indicated with the horizontal lines. (**C**) Summary of the data, current levels after two minutes of applying the various concentrations of the compounds were divided by basal current levels before the application of the drug. The data were fitted using the Hill1 equation with the Origin 8.0 software; data are plotted as mean ± SEM ($n = 6$ for each concentration for TRPV6 and $n = 8$–11 for each concentration for TRPV5). Inset shows the chemical formula for ZINC9155420. Asterisks denote chiral centers. (**D–E**) Representative traces for the effects of ZINC17988990 on (**D**) TRPV5 and (**E**) TRPV6 currents, the applications of various concentrations of ZINC17988990 are indicated with the horizontal lines. (**F**) Summary of the data, analyzed and plotted as in panel C ($n = 5$–15 for each concentration). Inset shows the chemical formula for ZINC17988990.

DOI: https://doi.org/10.7554/eLife.49572.002

The following source data and figure supplements are available for figure 1:

**Source data 1.** SBVS compound hits.

DOI: https://doi.org/10.7554/eLife.49572.004

**Source data 2.** In silico hits that had no effect on TRPV5 activity.

DOI: https://doi.org/10.7554/eLife.49572.005

**Source data 3.** ZINC9155420 derivatives that had no effect on TRPV5 activity.

DOI: https://doi.org/10.7554/eLife.49572.007

**Figure supplement 1.** TRPV5 inhibitor binding pocket from in silico screen.

DOI: https://doi.org/10.7554/eLife.49572.003

**Figure supplement 2.** Representative traces for tested compounds that had no effect on rbTRPV5 activity.

DOI: https://doi.org/10.7554/eLife.49572.006

compound also inhibited the hTRPV6 channel with a comparable $IC_{50}$ in the same system ($4.05 \pm 1.04$ µM, *Figure 1C*).

Five chemical analogs of ZINC9155420 were purchased based on vendor availability and price. These compounds were tested in the HEK293 cell expression system and four of these derivatives showed no effect on TRPV5 activity (*Figure 1—source data 3*). The fifth derivative, ZINC17988990, was found to be a potent inhibitor of TRPV5 mediated currents. This compound exhibited robust inhibition of rbTRPV5 with an $IC_{50}$ of $106 \pm 27$ nM while also showing marked selectivity for TRPV5 over TRPV6 (*Figure 1D–F*). Human TRPV6 was not inhibited by ZINC17988990 at concentrations up to 10 µM in the same system (*Figure 1E–F*). Additionally, ZINC17988990 had no activity on other TRP channels tested (TRPV1, TRPV3, TRPV4 and TRPM8) at low concentrations (1–3 µM), but did exhibit moderate inhibition on TRPV3 and TRPM8 at concentrations above 10 µM (*Figure 2*). Further characterization of ZINC17988990 revealed that it has comparable inhibition of the human isoform of TRPV5 ($IC_{50} = 177 \pm 47$ nM, *Figure 2—figure supplement 1*).

To confirm that $Ca^{2+}$ currents are inhibited similarly to monovalent currents, we performed fluorescence measurements in HEK293 cells transfected with the $Ca^{2+}$ sensor GCaMP6 and various TRP channels. We found 10 µM ZINC17988990 fully inhibited $Ca^{2+}$ signals elicited by application of external $Ca^{2+}$ in cells transfected with TRPV5, but it had no effect in cells expressing TRPV6 (*Figure 2—figure supplement 2A–B and H*), consistent with our monovalent current data. $Ca^{2+}$ signals elicited by agonists of TRPV1, TRPM8, TRPM3 and TRPV4 were not inhibited by 10 µM ZINC17988990 in cells transfected with the respective channel (*Figure 2—figure supplement 2D–H*). We also tested the effect of ZINC17988990 on rbTRPV5 expressed in Xenopus oocytes (*Figure 2—figure supplement 3*). Similar to our previous results with econazole (*Hughes et al., 2018b*), the concentrations required to achieve inhibition were higher in oocytes ($IC_{50} = 4.37 \pm 0.69$ µM) compared to HEK293 cells. As seen with our results in HEK cells, ZINC17988990 did not induce any inhibition of hTRPV6 in Xenopus oocytes up to 30 µM (*Figure 2—figure supplement 3*).

The other hit compound from the original screen, ZINC05626366, showed 73% inhibition of TRPV5 at 10 µM, but only 36% inhibition at 3 µM (*Figure 2—figure supplement 4*). Since this compound exhibited much lower potency than ZINC17988990, we have not characterized it further.

## Structural and functional characterization of novel TRPV5 inhibitors

In order to verify that the newly identified TRPV5 inhibitors, ZINC9155420 and ZINC17988990, bind to TRPV5, we utilized cryo-electron microscopy (cryo-EM) to solve the structure of rabbit TRPV5 in the presence of each compound. We were able to resolve nanodisc-reconstituted TRPV5 in the presence of 10 µM ZINC9155420 or ZINC17988990 to 4.3 Å and 3.8 Å, respectively (*Figure 3*, *Figure 3—figure supplements 1–4*, *Table 1*). These structures are consistent with the architecture of the previously reported structures of TRPV5 (*Hughes et al., 2018a*; *Hughes et al., 2018b*; *Dang et al., 2019*). The transmembrane domain (TMD) consists of transmembrane helices 1–4 (S1-S4) bundled in a voltage sensing-like domain with helices 5 and 6 (S5 and S6) domain swapped to create the pore of the channel. The intracellular portion of the protein is primarily composed of tightly packed ankyrin repeat domains (ARDs). Higher local resolution in the TMD of both structures allowed for the confident placement of side chains in this region when building the model (*Figure 3—figure supplements 1–4*). The lower resolution ARDs of both structures allowed for confident carbon backbone placement (*Figure 3—figure supplements 1–4*). At these resolutions, we were able to clearly identify that the ion conduction pathway for both inhibitor-bound structures are in non-conducting conformations (*Figure 3—figure supplement 5*). We were also able to identify at these resolutions non-protein densities in the TMD of both structures that we attributed to bound inhibitors (*Figure 3*). Due to the higher resolution of the ZINC17988990-bound TRPV5 structure coupled with the potent functional effect of this compound we have focused the majority of this investigation on the structural and functional effects of ZINC17988990 on TRPV5.

The ZINC17988990-bound TRPV5 structure revealed two densities that could be attributed to bound compounds (*Figure 3C–D*). One density identified in the ZINC17988990-bound TRPV5 structure is located between the intracellular S1-S4 bundle and the TRP helix and it is present in the sharpened map as well as both of the half-maps (*Figure 4A*, *Figure 4—figure supplement 1*). This S1-S4 bundle was resolved in this structure to a local resolution of 3.0–3.5 Å which allowed for confident backbone and side chain placement (*Figure 3—figure supplement 3*). Though the upper region of the S1-S4 pocket has been consistently occupied by lipids in the previous TRPV5 channel

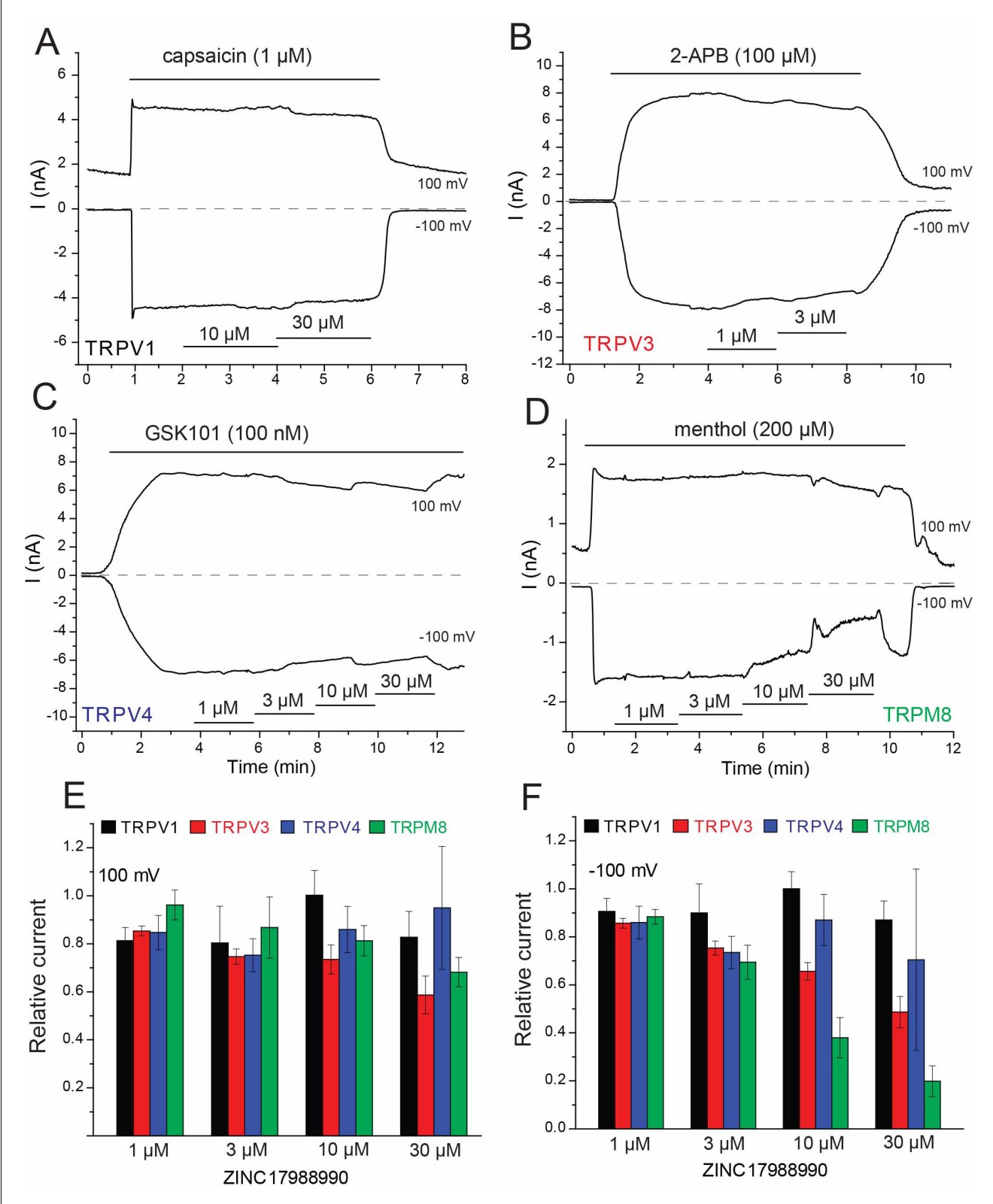

**Figure 2.** Effect of ZINC17988990 on other TRP channels. HEK293 cells were transfected with the mouse TRPV1, mouse TRPV3, the rat TRPV4 and the rat TRPM8 clones. Whole cell patch clamp experiments were performed as described in the methods section. (A–D) Representative traces for (A) TRPV1, (B) TRPV3, (C) TRPV4 and (D) TRPM8. The applications of the various channel agonists are shown with the horizontal lines above the current traces, the application of the different concentrations of ZINC17988990 are indicated by horizontal lines at the bottom. (E–F) Summary of the data at (E)

*Figure 2 continued on next page*

*Figure 2 continued*

100 mV and (F) −100 mV. Current levels after two minutes of applying the various concentrations of the compounds were divided by agonist-induced current levels before the application of the drugs. Data show mean ± SEM, n = 3–4 for TRPV1, n = 3–6 for TRPV3, n = 4–6 for TRPV4 and n = 5–7 for TRPM8.

DOI: https://doi.org/10.7554/eLife.49572.008

The following figure supplements are available for figure 2:

**Figure supplement 1.** The effect of ZINC17988990 on human TRPV5.

DOI: https://doi.org/10.7554/eLife.49572.009

**Figure supplement 2.** Effect of ZINC17988990 on TRP channel mediated calcium flow.

DOI: https://doi.org/10.7554/eLife.49572.010

**Figure supplement 3.** Effects of ZINC17988990 on rbTRPV5 and hTRPV6 expressed in Xenopus oocytes.

DOI: https://doi.org/10.7554/eLife.49572.011

**Figure supplement 4.** The effects of ZINC05626366 on rbTRPV5.

DOI: https://doi.org/10.7554/eLife.49572.012

structures (*Hughes et al., 2018a*; *Hughes et al., 2018b*; *Dang et al., 2019*), a density lower in the pocket has been identified for the first time in the ZINC17988990-bound structure (*Figure 4A*, *Figure 4—figure supplement 2*). In cryo-EM structures of other TRP channels, such as TRPM8 and TRPV6, densities attributed to small-molecule modulators were also identified in this S1-S4 pocket (*Figure 4—figure supplement 2*)(*Yin et al., 2019*; *Singh et al., 2018*). In the case of TRPV6, this was proposed in spite of lipid binding reported in the same area in previous structures (*Singh et al., 2018*; *McGoldrick et al., 2018*). This precedent, along with the absence of the lower density in previously published TRPV5 cryo-EM structures (*Hughes et al., 2018a*; *Hughes et al., 2018b*; *Dang et al., 2019*), has led us to attribute this density to bound ZINC17988990. The best fit for ZINC17988990 in this density is shown throughout the figures.

The residues that constitute this binding site in the S1-S4 bundle of the ZINC17988990-bound TRPV5 structure include E403, D406, Y415, Y467 and F468 (*Figure 4B*). Of these residues that contact the compound density in the EM map, only Y415 is not conserved between TRPV5 and TRPV6 (*Figure 4—figure supplement 3A*). In TRPV6, the residue at this position is a phenylalanine. In order to support ligand binding at this site, we tested the inhibition of ZINC17988990 on F468A, D406A and Y415F mutations of TRPV5 (*Figure 4C*). The F468A mutation created a non-functional channel, implying this residue is important for proper protein folding, trafficking to the membrane and/or activation by $PI(4,5)P_2$. The D406A mutation eliminated inhibition of the channel at low concentrations while maintaining the inhibitory effect at higher concentrations, resulting in an $IC_{50}$ of 3.68 ± 1.03 µM. The Y415F mutation also resulted in a reduced potency of ZINC17988990, resulting in an $IC_{50}$ of 0.644 ± 0.106 µM (Figure 4C). The substantially reduced inhibitory effects of these S1-S4 bundle mutant channels indicate that this compound exerts its inhibitory effect primarily by binding in this pocket. The D406A mutant did not reduce inhibition of TRPV5 by ZINC9155420, which appears to bind in a different area of the TMD (*Figure 3A–B*; also see following section), indeed this mutant required slightly lower concentrations of the drug than wild type (*Figure 4D*). This indicates that the D406A mutant did not reduce inhibition by ZINC17988990 non-specifically by increasing open state stability of the channel, but rather by specifically interfering drug binding.

The second density located near the SBVS pocket appeared to be bound at the interface of the S4-S5 linker of one monomer and the S6 helix of an adjacent monomer (*Figure 4—figure supplement 4*). This region of TRPV5 has a local resolution of 3.0–3.5 Å which allowed for unambiguous side chain and backbone placement (*Figure 3—figure supplements 3–4*). It should be noted that this density, though present in the sharpened cryo-EM density map, was not consistently visible in the half-maps and therefore could be attributed to either low occupancy inhibitor binding or noise (*Figure 4—figure supplement 5*). To illustrate how the inhibitor could fit in the density, one proposed conformation of ZINC17988990 is shown throughout the figures. To further investigate the contribution of this pocket to channel inhibition we tested the effect of ZINC17988990 on M491A mutated TRPV5 (*Figure 4—figure supplement 4*). We observed only a small decrease in the inhibitory potency of ZINC17988990 on this mutant ($IC_{50}$ = 0.258 ± 0.093 µM; *Figure 4—figure supplement 4*). Furthermore, inhibition of the M491A-D406A double mutant by ZINC17988990 was

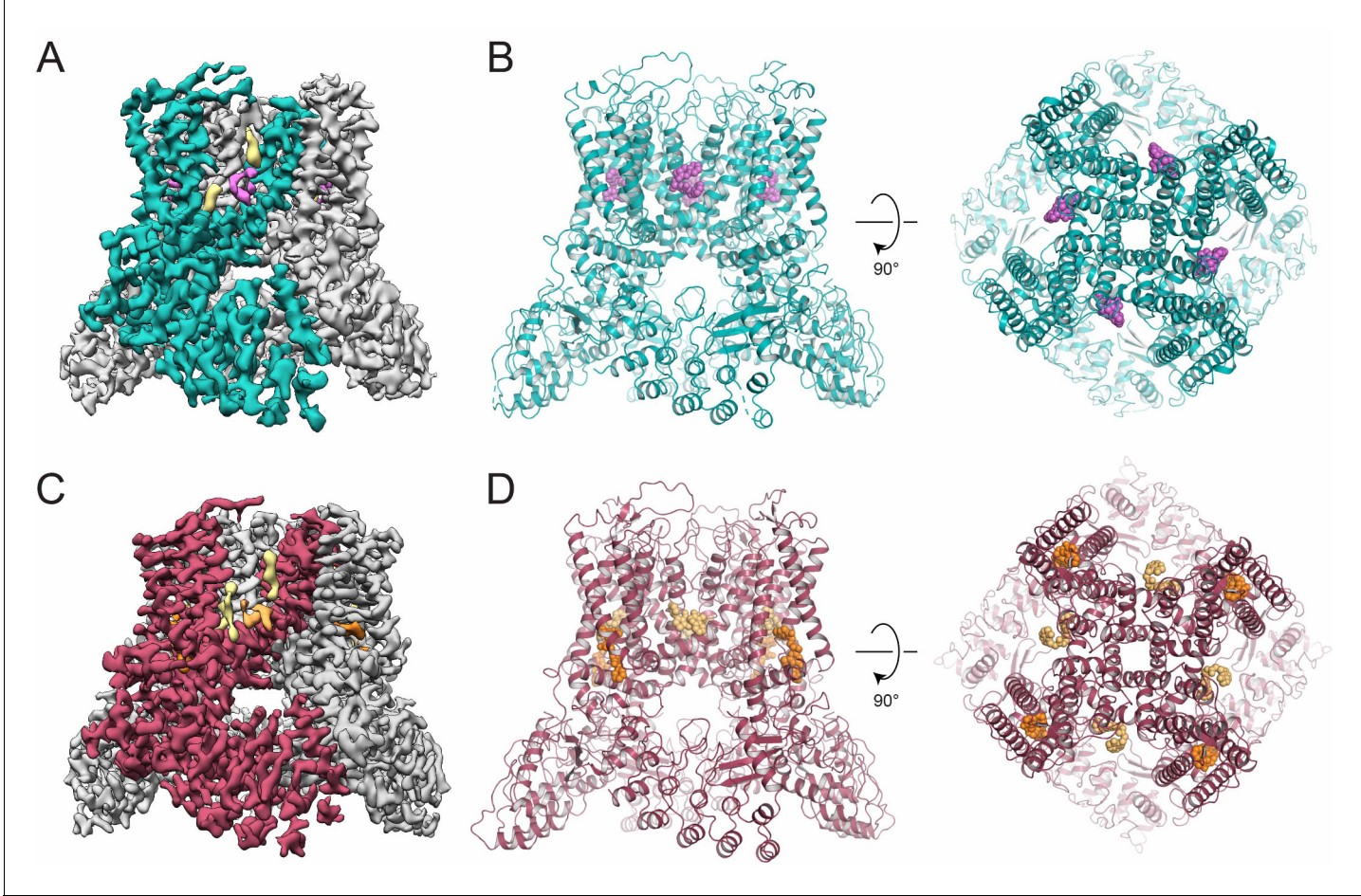

**Figure 3.** Inhibitor-bound TRPV5 cryo-EM structures. (**A**) Cryo-EM density map of ZINC9155420-bound TRPV5 in nanodiscs. A single TRPV5 monomer is colored in teal while the remaining are shown in gray. Density attributed to bound ZINC9155420 is shown in pink and lipids are shown in khaki. (**B**) Atomic model of ZINC9155420-bound TRPV5 in nanodiscs. One potential pose of the bound ZINC9155420 molecule is shown as pink spheres. (**C**) Cryo-EM density map of ZINC17988990-bound TRPV5 in nanodiscs. A single TRPV5 monomer is colored in red while the remaining are shown in gray. Densities attributed to bound ZINC17988990 are shown in orange and light orange. Lipids are shown in khaki. (**D**) Atomic model of ZINC17988990-bound TRPV5 in nanodiscs. Potential poses for bound ZINC17988990 in each binding location are shown as orange spheres.

DOI: https://doi.org/10.7554/eLife.49572.013

The following figure supplements are available for figure 3:

**Figure supplement 1.** Cryo-EM overview of ZINC9155420-bound TRPV5 in nanodiscs.
DOI: https://doi.org/10.7554/eLife.49572.014

**Figure supplement 2.** Data quality of ZINC9155420-bound TRPV5 in nanodiscs.
DOI: https://doi.org/10.7554/eLife.49572.015

**Figure supplement 3.** Cryo-EM overview of ZINC17988990-bound TRPV5 in nanodiscs.
DOI: https://doi.org/10.7554/eLife.49572.016

**Figure supplement 4.** Data quality of ZINC17988990-bound TRPV5 in nanodiscs.
DOI: https://doi.org/10.7554/eLife.49572.017

**Figure supplement 5.** Inhibition of TRPV5 by novel compounds.
DOI: https://doi.org/10.7554/eLife.49572.018

indistinguishable from the D406A S1-S4 bundle mutant (*Figure 4—figure supplement 4*). These data show that the S4-S5 linker is unlikely to play a major role in ZINC17988990-mediated inhibition.

## Insights into the mechanism of specific TRPV5 inhibition

This ZINC17988990-inhibited TRPV5 structure, in combination with the other previously solved TRPV5 structures, provides insight as to how binding in the S1-S4 bundle could lead to channel

**Table 1.** Cryo-EM data collection and model statistics

| | ZINC9155420-Bound TRPV5 in nanodiscs (PDB: 6PBF, EMB-20292) | ZINC17988990-bound TRPV5 in nanodiscs (PDB: 6PBE, EMB-20291) |
| --- | --- | --- |
| **Data collection and processing** | | |
| Magnification | ~45,500 | ~45,500 |
| Voltage (kV) | 300 | 300 |
| Defocus range (μm) | 1.0–2.5 | 0.8–2.5 |
| Pixel size (Å) | 1.064 | 1.064 |
| Symmetry imposed | C4 | C4 |
| Initial particle images (no.) | 510,500 | 670,057 |
| Final particle images (no.) | 21,802 | 98,516 |
| Map resolution (Å) FSC threshold | 4.2 0.143 | 3.78 0.143 |
| Map resolution range (Å) | 3.5–5.5 | 3.0–5.0 |
| **Refinement** | | |
| Model resolution cut-off (Å) FSC threshold | 4.3 0.143 | 3.78 0.143 |
| Map sharpening $B$ factor (Å$^2$) | −330 | −253 |
| Model composition | 0 | 0 |
| Nonhydrogen atoms | 2392 | 2416 |
| Protein residues | 4 | 4 |
| Ligands | | |
| R.m.s. deviations | 0.005 | 0.006 |
| Bond lengths (Å) | 0.897 | 0.935 |
| Bond angles (°) | | |
| Validation | 1.70 | 1.72 |
| MolProbity score | 5.02 | 3.88 |
| Clashscore | 0.22 | 1.18 |
| Poor rotamers (%) | | |
| Ramachandran plot | 93.20 | 91.83 |
| Favored (%) | 6.80 | 8.17 |
| Allowed (%) | 0.00 | 0.00 |
| Disallowed (%) | | |

DOI: https://doi.org/10.7554/eLife.49572.019

inhibition. The binding site of the PI(4,5)P$_2$ head group, a well characterized endogenous activator of TRPV5, does not overlap with this identified inhibitor binding site, which implies that the mechanism of inhibition for this novel specific compound is not to directly compete for binding with PI(4,5)P$_2$. Rather, it is possible that conformational changes due to inhibitor binding could lock the channel in a state that would not allow for PI(4,5)P$_2$-mediated activation.

Large movements of the TRPV5 channel have been shown to occur during PI(4,5)P$_2$ induced channel opening (*Hughes et al., 2018a*). Specifically, the rearrangement of the lower S6 helix caused by the phosphate head group binding to R584 causes global shifts in the protein from the lipid-bound apo state (PDB: 6DMR) to the opened PI(4,5)P$_2$-bound state (PDB: 6DMU). While the ZINC17988990-bound TRPV5 structure is similar to the lipid-bound apo structure, there are several key differences when comparing this inhibited state to the lipid-bound apo state.

In particular, when aligned based on the pore dimer (all atom alignment of M497-R584 of opposite chains using PyMOL) the S1-S4 bundle of the ZINC17988990-bound structure appears to have undergone a shift from the lipid-bound apo structure to accommodate the binding of ZINC17988990. Specifically, the S2 and S3 helices of the ZINC17988990-bound TRPV5 have moved

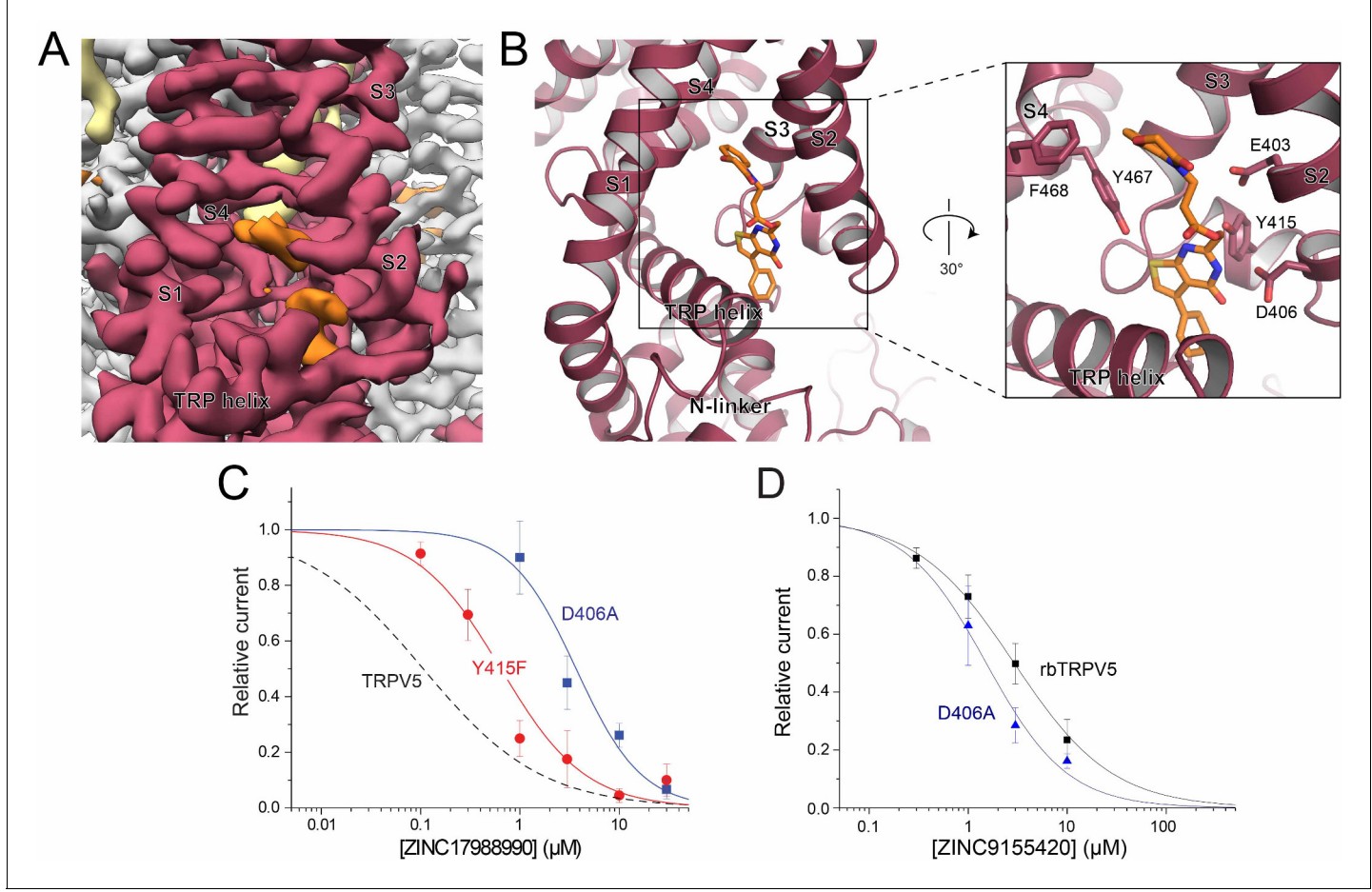

**Figure 4.** ZINC17988990 binding in the S1-S4 bundle. (A) Cryo-EM density map of the ZINC17988990 binding pocket in the S1-S4 bundle. Density attributed to ZINC17988990 is shown in orange. Lipids are colored in khaki. The TRPV5 protein is depicted with one monomer in red and three in gray. (B) (left) Atomic model of ZINC17988990 binding pocket in the lower portion of the S1-S4 bundle. (right) Zoomed in view of the binding pocket with residues that could constitute binding labeled and shown as sticks. One potential pose of the ZINC17988990 molecule is shown as orange sticks. (C) Summary of the effects of ZINC17988990 on wild type TRPV5 (dashed line, replotted from *Figure 1F*) and the Y415F (red) and D406A (blue) mutants analyzed and plotted as in *Figure 1*. $n$ = 3–9 and 5–11 for each concentration for D406 and Y415F, respectively. (D) Summary of whole cell patch clamp experiments in HEK293 cells expressing wild type rbTRPV5 (black, replotted from *Figure 1C*) and D406A rbTRPV5 (blue, $n$ = 6 for each concentration) in the presence of ZINC9155420 analyzed and plotted as in *Figure 1*.

DOI: https://doi.org/10.7554/eLife.49572.020

The following figure supplements are available for figure 4:

**Figure supplement 1.** Cryo-EM half maps of intracellular S1-S4 bundle binding pocket.

DOI: https://doi.org/10.7554/eLife.49572.021

**Figure supplement 2.** Ligand binding in the intracellular S1-S4 bundle.

DOI: https://doi.org/10.7554/eLife.49572.022

**Figure supplement 3.** Select sequence alignments.

DOI: https://doi.org/10.7554/eLife.49572.023

**Figure supplement 4.** ZINC17988990 interaction with the S4-S5 linker.

DOI: https://doi.org/10.7554/eLife.49572.024

**Figure supplement 5.** Cryo-EM half maps of the S4-S5 binding pocket.

DOI: https://doi.org/10.7554/eLife.49572.025

toward the ZINC17988990 density, which creates a tighter space with in the intracellular section of the S1-S4 bundle compared to the lipid-bound apo TRPV5 structure (*Figure 5A*). Residues E403, D406 and Y415 appear to move 1–2.5 Å (as measured at the Cα) to accommodate compound

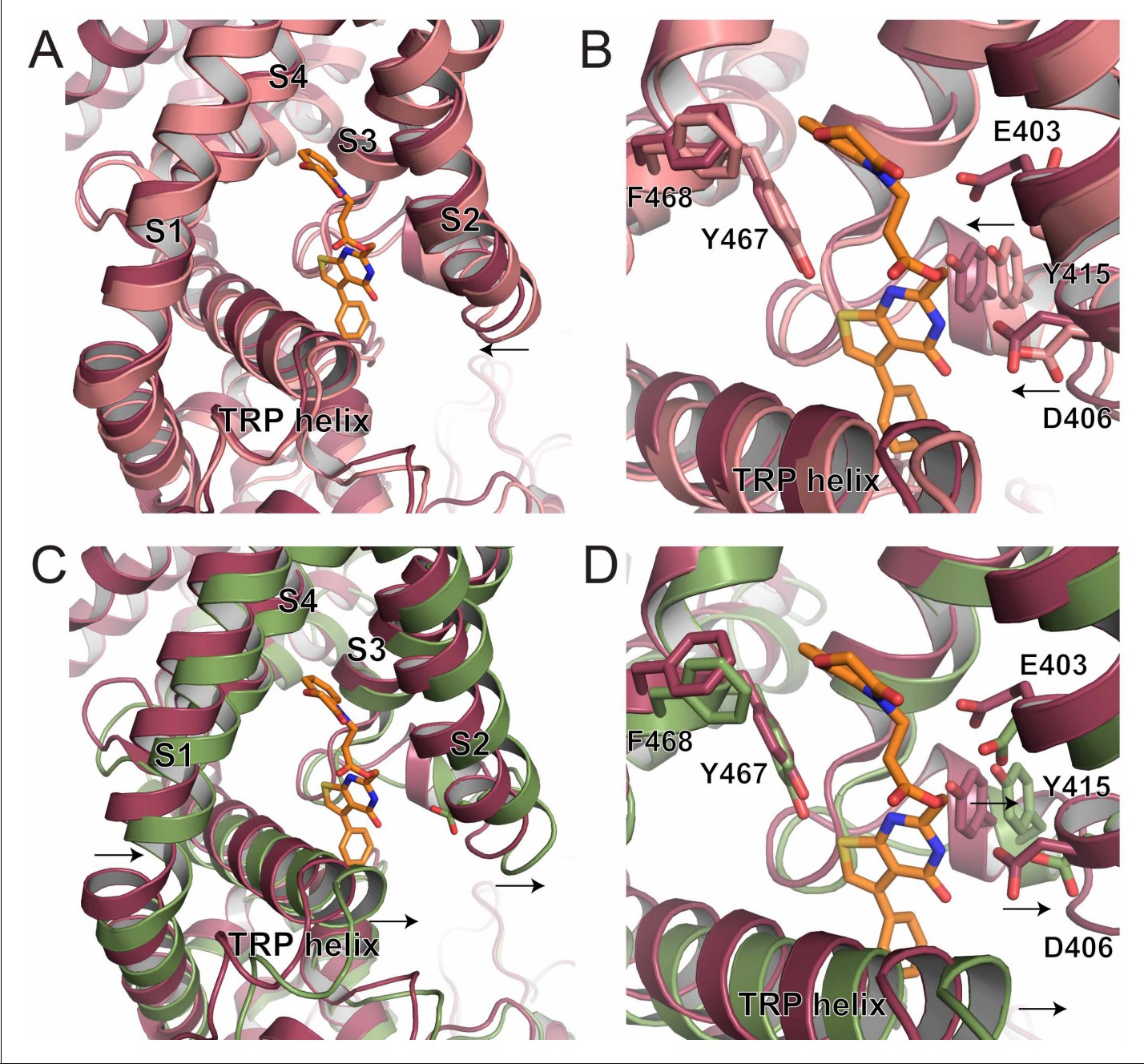

**Figure 5.** S1-S4 bundle mediated inhibition of TRPV5. (**A**) Overlay of the lower S1-S4 bundle in ZINC17988990-bound (red) and lipid-bound (salmon) TRPV5. The arrow indicates the movement attributed to ZINC17988990 binding. (**B**) Overlay of the inhibitor binding pocket of ZINC17988990-bound (red) and lipid-bound (salmon) TRPV5. The arrows indicate the movement attributed to ZINC17988990 binding. Residues that could constitute ZINC17988990 binding are labeled and shown as sticks. (**C**) Overlay of the lower S1-S4 bundle in ZINC17988990-bound (red) and PI(4,5)P$_2$-bound (green) TRPV5. The arrows indicate movement necessary for PI(4,5)P$_2$ activation. (**D**) Overlay of the inhibitor binding pocket of ZINC17988990-bound (red) and PI(4,5)P$_2$-bound (green) TRPV5. The arrows indicate movement necessary for PI(4,5)P$_2$ activation. Labels of residues that could constitute ZINC17988990 binding are labeled and shown as sticks.

DOI: https://doi.org/10.7554/eLife.49572.026

The following figure supplements are available for figure 5:

**Figure supplement 1.** The effect of reducing PI(4,5)P$_2$ levels with a voltage sensitive phosphatase on wild type and mutant TRPV5 channels.
DOI: https://doi.org/10.7554/eLife.49572.027
**Figure supplement 2.** Structural divergence between TRPV5 and TRPV6 at the S2-S3 linker.
DOI: https://doi.org/10.7554/eLife.49572.028

binding, while F468 and Y467 do not appear to undergo significant conformational rearrangement (<1 Å movement at the Cα; *Figure 5B*).

Conformational changes undergone in the S1-S4 bundle upon PI(4,5)P$_2$ binding include movement away from the pore axis of all four helices, with the largest movements in the lower halves of the S1 and S2 helices (aligned as described above, *Figure 5C–D*). Additionally, a counterclockwise rotation of the bundle and a pivot of the TRP helix occur accommodate the helix transition at the base of the ion conduction pore (*Figure 5C–D*). Based on the positioning of the ZINC17988990 density, these shifts and rotations would be limited by the interactions of ZINC17988990 with D406 and Y415 (*Figure 5D*). These stabilizing interactions could act to lock this inhibited conformation of the channel in place and not allow for the shifts in the S1-S4 bundle that are necessary for PI(4,5)P$_2$ binding and subsequent activation. Consistent with this idea, we found that the D406A mutation slightly, but significantly reduced the inhibition of TRPV5 evoked by decreasing PI(4,5)P$_2$ using the voltage dependent lipid phosphatase drVSP (*Figure 5—figure supplement 1A,B*).

To shed light on the mechanism of selectivity of ZINC17988990 we compared this S1-S4 bundle binding site between TRP family channels (*Figure 4—figure supplement 3A*). Specifically, a comparison of this region in TRPV5 and TRPV6 shows very high sequence homology and therefore the selectivity of ZINC17988990 is not likely due to a difference in sequence in this region (*Figure 4—figure supplement 3A*). Rather, it appears that the structural difference between TRPV5 and TRPV6 in this region may play a role. The 16 amino acid linker between the S2 and S3 helices of TRPV5 has consistently been resolved as a stable short helix while in the many cryo-EM and X-ray crystallography structures of TRPV6, this linker has not been resolved (*Singh et al., 2018*; *McGoldrick et al., 2018*; *Saotome et al., 2016*) (*Figure 5—figure supplement 2*). This implies that this region is extremely flexible in TRPV6 while it is consistently more stable in TRPV5 (*Hughes et al., 2018a*; *Hughes et al., 2018b*; *Gao et al., 2016*) (*Figure 5—figure supplement 2*). Both Y415 and D406, which make contacts with the drug density in the ZINC17988990-bound TRPV5 structure, are unresolved in all reported TRPV6 structures. The lack of stability in this region in TRPV6 may not allow for those residues to establish interactions with ZINC17988990, preventing effective inhibition. This structural difference could be due to small differences in sequence in the regions that stabilize the S2-S3 linker in TRPV5, such as the TRP domain and the N-linker.

## Nonspecific inhibition of TRPV5

While severely limited by resolution, the ZINC9155420-bound TRPV5 structure was still able to give us some insights on the interactions between this novel nonspecific inhibitor and the TRPV5 channel. In the ZINC9155420-bound TRPV5 structure, the density attributed to bound inhibitor is located at the interface between the S4-S5 linker of one monomer and the S6 helix of an adjacent monomer (*Figure 6A–C*). The local resolution of this region is ~3.5 Å which has allowed for confident side chain and backbone placement (*Figure 3—figure supplements 1–2*). This inhibitor density is found in both half maps and the sharpened density map of the ZINC9155420-bound structure and has not been seen in other TRPV5 cryo-EM structures to date (*Hughes et al., 2018a*; *Hughes et al., 2018b*; *Dang et al., 2019*) (*Figure 4—figure supplement 5*). Two additional densities appear to bracket the compound and, given their characteristic elongated shapes and their presence in previously reported TRPV5 cryo-EM structures (*Hughes et al., 2018a*; *Dang et al., 2019*), they have been assigned to lipid molecules which may be involved in compound binding (*Figure 6A*). ZINC9155420 was purchased as a racemic mixture and therefore the functional and structural screens were not able to determine which of the stereoisomers exhibited the inhibitory effect. Additionally, at this resolution the inhibitor density could accommodate multiple orientations of the compound, and limitations in docking scoring functions do not allow us to chose one over another. One potential fit of the *S*-enantiomer of ZINC9155420 is shown throughout the figures.

It should be noted that the low particle number (<5% of the initial particles) in the final ZINC9155420-bound TRPV5 structure was not due to problems of occupancy, as a density that we attributed to bound ZINC9155420 was present in the preliminary 3D refinement. Rather, the particles selected were the highest resolution particles in the dataset. It is possible that in the presence of ZINC9155420, TRPV5 adopts multiple transition states which could have resulted in the small number of particles in the final structure, though none of the classes discarded during processing were of high enough resolution to clearly identify alternative states.

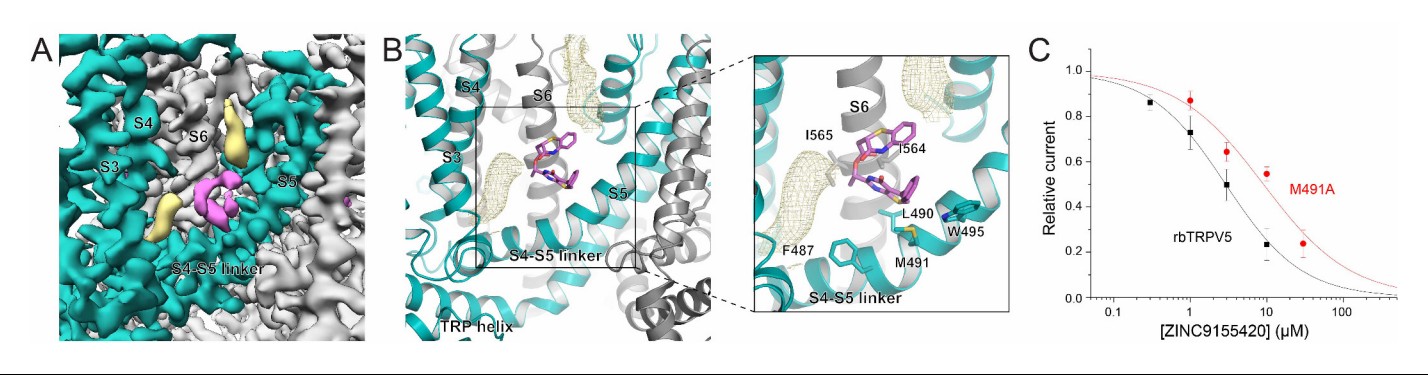

**Figure 6.** ZINC9155420 binding and inhibition. (**A**) Cryo-EM density map of the ZINC9155420 binding pocket. Density attributed to ZINC9155420 is shown in pink. Lipids are colored in khaki. The TRPV5 protein is depicted with one monomer in teal and three in gray. (**B**) (left) Atomic model of the ZINC9155420 binding pocket. (right) Zoomed in view of the ZINC9155420 binding pocket with residues that could constitute binding labeled and shown as sticks. One potential pose of the ZINC9155420 molecule is shown as pink sticks and densities attributed to lipids are shown as khaki mesh. (**C**) Summary of the effects of ZINC9155420 on wildtype TRPV5 (black, replotted from *Figure 1C*) and M491A TRPV5 (red, *n* = 4–7) analyzed and plotted as in *Figure 1*.

DOI: https://doi.org/10.7554/eLife.49572.029

The following figure supplement is available for figure 6:

**Figure supplement 1.** Predicted vs. observed binding.
DOI: https://doi.org/10.7554/eLife.49572.030

In order to functionally support compound binding at the S4-S5 pocket, we tested the inhibition of ZINC9155420 on the M491A mutant of rbTRPV5 (*Figure 6C*). This mutation resulted in a right shift of the inhibition curve (IC$_{50}$ = 9.08 ± 1.45 µM) which suggests that M491 on the S4-S5 linker is involved in ZINC9155420 binding and inhibition. The lack of selectivity of ZINC9155420 may be due to the extremely high level of sequence homology of this region between TRP family channels (*Figure 4—figure supplement 3B*). This S4-S5 linker has been shown to be critical for gating of several TRP channels and may be evolutionarily conserved due to its essential role in channel function (*Hofmann et al., 2017*).

Unlike the ZINC17988990-bound TRPV5 structure, the S1-S4 bundle appears to be occupied by a lipid as seen in other TRPV channel structures (*Figure 4—figure supplement 2*). As mentioned earlier, we tested the effect of ZINC9155420 on the D406A mutant of TRPV5 in the S1-S4 bundle binding site; this mutation caused a small left shift of the inhibition curve for ZINC9155420 (IC$_{50}$ = 1.52 ± 0.38 µM, *Figure 4D*). This implies that this S1-S4 binding site is unlikely to be involved in ZINC9155420 inhibition of TRPV5.

Neither inhibitor characterized in this investigation appeared to bind at the econazole binding pocket, originally used as the site for the SBVS study. Though this S4-S5 binding site is near the econazole pocket there are no residues that overlap with the cryo-EM attributed binding sites (within 4.5 Å of docked poses at the econazole binding site, *Figure 6—figure supplement 1*). Rather, in both inhibitor-bound structures, this econazole pocket appears to be occupied by a tightly bound lipid molecule. This density in the inhibitor-bound TRPV5 structures appears to closely resemble in size and shape the lipid density in the lipid-bound TRPV5 structure rather than the econazole density seen in the econazole-bound TRPV5 structure. The competition between a lipid and econazole for the site we utilized for SBVS was unexpected. Notably, a post hoc SBVS screen using (a) the ZINC9155420-bound TRPV5 cryo-EM atomic model with a lipid molecule in the econazole pocket and the other identified lipid in the vicinity, (b) the same ZINC15 compound library we used in the original screen, and (c) a screening grid centered on ZINC9155420, identified ZINC9155420 and ZINC17988990, as well as other derivatives of these two molecules, among the top-scored compounds. Therefore, while SBVS protocols utilize predefined binding sites, and we did not foresee lipid occupancy of the initially used econazole site, the retrospective identification of the same novel TRPV5 inhibitors characterized here if one used the correct binding pocket gives confidence in the approach used.

## Discussion

Here we have reported three novel TRPV5 inhibitors. One of these compounds, ZINC17988990 is specific for TRPV5 over its close homologue TRPV6 as well as other TRP family channels. ZINC17988990 inhibits TRPV5 with higher affinity and higher specificity than previously described compounds, and therefore will be a highly valuable tool in future experiments to study TRPV5.

Structural and functional characterization of two of these compounds identified two novel inhibitor binding sites for the TRPV5 channel. One binding site was identified at the interface between lipids, the S4-S5 linker of one monomer and the S6 helix of an adjacent monomer and appears to play a role in ZINC9155420-mediated inhibition. The other site identified in the ZINC17988990-bound TRPV5 structure occupies the intracellular half of the S1-S4 bundle. This binding site partially overlaps with modulators seen in the recent TRPM8 structures as well as the 2-APB binding site reported for TRPV6 (*Figure 4—figure supplement 2*) (*Yin et al., 2019*; *Singh et al., 2018*). 2-APB is a small-molecule inhibitor of TRPV6 which acts as an activator for other TRPV subfamily channels while having no effect on TRPV5 function (*Singh et al., 2018*; *Hu et al., 2004*; *Kovacs et al., 2012*). The specificity of both 2-APB and ZINC17988990 in spite of the high sequence homology in this pocket between TRPV5 and TRPV6 may be explained by the structural divergence of these channels in this region.

An analysis of other TRPV5 cryo-EM structures has provided insight into a potential mechanism of action of the specific inhibitor by locking TRPV5 in closed states that may not allow for the conformational changes necessary for PI(4,5)P$_2$ binding and subsequent activation. Specifically, the binding of key residues such as D406 and Y415 in the S1-S4 pocket, that have not been seen to interact with any non-protein density in previous structures, could limit movement in this region and confer inhibition.

The success of this multidisciplinary approach has allowed for the discovery and characterization of two novel inhibitors of TRPV5 that have the potential to be optimized for higher potency and selectivity. Moreover, the identification of a binding site that allows for specificity of TRPV5 over TRPV6 in spite of the high sequence homology now allows for a more controlled structure-based drug discovery effort at this pocket that has the potential to reveal additional specific small molecule modulators of TRPV5. Overall, this study has laid the groundwork for the development of more potent and specific inhibitors of TRPV5.

## Materials and methods

### Structure-based virtual screening

The 3.9 Å resolution cryo-electron microscopy structure of the lipid-bound TRPV5 tetramer (PDB ID: 6DMR) was used for a first SBVS study. The default protocol of the Protein Prep Wizard in the Schrodinger suite 2018-1 (*Sastry et al., 2013*) was followed to prepare the protein for docking. Specifically, missing hydrogen atoms were added, side chain protonation states were assigned, and energy minimization was carried out using the OPLS3 force-field (*Harder et al., 2016*). Since the four monomers of TRPV5 are identical in the starting TRPV5 tetramer structure, monomer A was chosen as a representative structure to build a docking grid using the Schrodinger's Receptor Grid generation tool, with the grid centered at the lipid molecule's center in the binding pocket of the protein monomer. The box dimensions for the inner and outer grid boxes were 10 Å x 10 Å x 10 Å, and 30 Å x 30 Å x 30 Å, respectively.

The ZINC15 (http://zinc15.docking.org/) 'Drugs Now' subset consisting of 11,768,355 compounds as of March 27, 2018 was downloaded and prepared for docking using the default procedure in the Schrodinger's Ligprep utility. Following ligand preparation and protein grid generation, Glide was used to dock molecules of the 'Drugs Now' subset in two steps. First, Glide standard precision (SP) was used to preliminarily screen the ligand library (*Halgren et al., 2004*; *Friesner et al., 2004*). The top-ranked molecules based on Glide SP reached a docking score plateau at −7.0. Secondly, the resulting 197,820 molecules were subjected to a Glide's extra precision (XP) refinement (*Friesner et al., 2006*). The top-ranked 100 molecules resulting from this refinement were then grouped into 65 clusters based on their chemical similarity calculated using Tanimoto similarity scores of ECFP4 binary fingerprints in the Schrodinger's Canvas program (*Duan et al., 2010*; *Sastry et al., 2010*). Notably, none of them exhibited significant similarity (defined as Tanimoto

similarity score larger than 0.4) with known inhibitors of TRPV5 (*Landowski et al., 2011*; *Simonin et al., 2015*).

A second SBVS was carried out using the ZINC9155420-bound TRPV5 cryo-EM structure. The structure was prepared for docking following the same procedure as described above, and the grid was centered at the ligand's (ZINC9155420) center in the binding pocket of monomer A inferred by the cryo-EM density map. Both lipid molecules assumed to occupy the additional densities identified near the compound were kept in this SBVS study. Docking was carried out following the same protocol described above and using the same compound library. Glide SP docking score plateaued at around −5.5, and the resulting 860,000 molecules were selected for Glide XP docking refinement. Notably, the ZINC9155420 and ZINC17988990 compounds, as well as other analogs in the library (as defined by Tanimoto index >0.4), were identified among the Glide XP top-scoring ligands.

## Electrophysiology

Human Embryonic Kidney 293 (HEK293) cells were purchased from American Type Culture Collection (ATCC), Manassas, VA, (catalogue number CRL-1573), RRID:CVCL_0045; cell identity was verified by STR analysis. Passage number of the cells was monitored, and cells were used up to passage number 25–30, when a new batch with low passage number was thawed; cells were tested for the lack of mycoplasma infection. Cells were cultured in MEM supplemented with 10% FBS in 5% $CO_2$ at 37°C, and they were transfected using the Effectene reagent (Qiagen). Measurements were performed 48–72 hr after transfection as described earlier (*Hughes et al., 2018b*). Briefly, currents were measured with an Axopatch 200B amplifier using a ramp protocol from −100 mV to 100 mV applied every second. The extracellular solution contained 142 mM LiCl, 1 mM $MgCl_2$, 10 mM HEPES, and 10 mM glucose, pH 7.4. Monovalent currents were initiated by the same solution containing 1 mM EGTA but no $MgCl_2$. The intracellular pipette solution contained 140 mM K-gluconate, 10 mM HEPES, 5 mM EGTA, 2 mM MgCl, and 2 mM ATP, pH 7.3. For both HEK cells and oocytes, we used a LiCl-based extracellular solution, because in this solution, removal of extracellular $Mg^{2+}$ and $Ca^{2+}$ minimizes the development of endogenous currents in nontransfected cells, compared to $Na^+$ or $K^+$ based solutions. For experiments with the danio rerio voltage sensitive phosphatase (drVSP) the membrane potential was clamped at −100 mV, and during the application of the $Ca^{2+}$ and $Mg^{2+}$ free solution four consecutive depolarizing pulses to +100 mV were applied for 0.1 s, 0.3 s, 1 s and 3 s. Currents were filtered at 5 kHz, and digitized through a Digidata 1440A interface. Data were collected and analyzed with the pClamp 10.6 acquisition software (RRID:SCR_011323; Molecular Devices, Sunnyvale, CA) and further analyzed and plotted with the Origin 8.0 software (Microcal Software Inc, Northampton, MA, USA; RRID:SCR_002815).

Two electrode voltage clamp (TEVC) experiments in Xenopus oocytes were performed as described earlier (*Hughes et al., 2018b*). Briefly, oocytes were digested using 0.2 mg/ml collagenase (Sigma) in a solution containing 82.5 mM NaCl, 2 mM KCl, 1 mM $MgCl_2$, and 5 mM HEPES, pH 7.4 (OR2) overnight for ~16 hr at 18°C. Defolliculated oocytes were kept at 18°C in OR2 solution supplemented with 1% penicillin/streptomycin (Mediatech) and 1.8 mM $CaCl_2$. cRNA was microinjected into each oocyte using a nanoliter injector system (World Precision Instruments). The experiments were performed 48–72 hr after injection. TEVC measurements were performed in a solution containing 97 mM NaCl, 2 mM KCl, 1 μM $MgCl_2$, and 5 mM HEPES, pH 7.4, monovalent currents were initiated by a solution containing 96 mM LiCl, 1 mM EGTA, and 5 mM HEPES, pH 7.4. Currents were measured with a ramp protocol from −100 to 100 mV applied every 500 ms, preceded by a step to −100 mV for 50 ms.

All tested compounds were dissolved in DMSO as 10 mM stock solutions. The stock was further diluted in the experimental buffer. Three compounds did not dissolve at this concentration in DMSO see *Figure 1—source data 2*, those compounds were not tested on TRPV5 inhibition. ZINC9155420 was unstable as a frozen DMSO stock, it lost activity after a couple of days, therefore we made a fresh DMSO stock before each experiment.

## Intracellular $Ca^{2+}$ measurements

HEK293 cells were transfected with rTRPV1, rTRPV4, rbTRPV5, hTRPV6, mTRPM3α2, rTRPM8 or pcDNA3.0 and with GCaMP6 (gift from Dr. Lawrence Larry Gaspers, NJMS) in a ratio 0.5:0.1 μg using the Effectene reagent (Qiagen). After 24 hr transfected cells were plated in poly-D-lysine

coated black-wall clear-bottom 96-well plates and measurements were performed 48 hr after plating. Thirty minutes before experiments the MEM media on the cells transfected with TRPV1, TRPV4, TRPM3 and TRPM8 was replaced with solution containing (in mM) 137 NaCl, 5 KCl, 1 MgCl$_2$, 2 CaCl$_2$, 10 HEPES and 10 glucose, pH 7.4. and for the cells transfected with TRPV5 and TRPV6 MEM was replaced with a nominally divalent free (NDF) solution containing (in mM) 137 NaCl, 5 KCl, 10 HEPES and 10 glucose, pH 7.4 and the plate was kept at 25℃. GCaMP6 signals were measured at excitation wavelengths 485 nm and fluorescence emission was collected at 525 nm using a 96-well plate reader Flexstation three with rapid well injection (Molecular Devices). Sampling interval was 0.86 s and four parallel reads were performed for each condition; the four reads were averaged, and they were treated as one data point for statistical analysis.

## Protein preparation

TRPV5 was expressed, purified and reconstituted into lipid nanodiscs as previously reported (*Hughes et al., 2018a*). In brief, rabbit TRPV5 was expressed in *S. cerevisiae* with a C-terminal 1D4 epitope tag. Yeast cells were lysed using a microfluidizer and the membranes were isolated via centrifugation at 3,000 g, 14,000 g and 100,000 g. The membranes were solubilized in buffer containing 20 mM HEPES, pH 8.0, 150 mM NaCl, 10% glycerol, 2 mM TCEP, 1 mM PMSF and 0.87 mM LMNG. Insoluble material was removed via ultracentrifugation. The solubilized TRPV5 was incubated with 1D4 conjugated CnBr-activated Sepharose 4B beads and washed with buffer containing 20 mM HEPES, pH 8.0, 150 mM NaCl, 2 mM TCEP, and 0.064 mM DMNG. Bound TRPV5 was eluted from the 1D4 beads with 20 mM HEPES, pH 8.0, 150 mM NaCl, 2 mM TCEP, 0.064 mM DMNG and 3 mg/mL 1D4 peptide. TRPV5 was then reconstituted into nanodiscs by combining TRPV5, MSP2N2 and soy polar lipids in a molar ratio of 1:1:200. Purified MSP2N2 was prepared using previously published methods. To prepare the lipids, soy polar lipid extract (Avanti Polar Lipids, Inc) were dried under nitrogen flow for 3 hr and resuspended in buffer containing 20 mM HEPES, pH 8.0, 150 mM NaCl, 2 mM TCEP and DMNG detergent at a molar ratio of 1:2.5 (lipids: DMNG). The TRPV5, MSP2N2 and lipids were incubated for 30mins then Bio-Beads (Bio-Rad, Bio-Beads SM-2 Absorbent Media) were added to the mixture for 1 hr. Fresh Bio-Beads were added and the mixture was allowed to incubate overnight with rotation. The reconstituted nanodiscs were then passed over Superose 6 (GE Healthcare) size-exclusion column equilibrated with 20 mM HEPES, pH 8.0, 150 mM NaCl, and 2 mM TCEP. The nanodisc reconstituted TRPV5 was concentrated to 2.2 mg/mL and incubated with 10 μM ZINC9155420 or 10 μM of ZINC17988990 prior to vitrification.

## Cryo-EM data acquisition

Both TRPV5 nanodisc samples were incubated with 3 mM F-Fos Choline eight immediately before vitrification. Samples were double blotted on 1.2/1.3 Quantifoil Holey Carbon grids (Quantifoil Micro Tools) with 3.5 μL of sample per blot. The samples were then vitrified in liquid ethane using a Vitrobot (Thermo Scientific). Both samples were imaged on a Titan Krios 300 kV electron microscope equipped with a Gatan K2 Summit direct electron detector. 40 frame movies were collected with a nominal dose of 40 e⁻/Å$^2$, a dose rate of ~5 e⁻/s/phys. pixel and a super resolution pixel size of 0.532 Å. The ZINC9155420-bound TRPV5 data set was collected in a defocus range of 1.0–2.5 μm under focus. The 10 μM of ZINC17988990-bound TRPV5 data set was collected between 0.8–1.8 μm under focus.

## Data processing

### ZINC9155420-bound TRPV5 structure determination

Motioncor2 was used to align each of the 2278 movies and adjust for beam induced motion (*Zheng et al., 2017*). CTF estimation was performed using GCTF on the summed movies (*Zhang, 2016*). Subsequent data processing was performed on the dose weighted micrographs in RELION (*Afonine et al., 2018*; *Kimanius et al., 2016*; *Scheres, 2016*) unless otherwise specified. 1838 particles were manually picked and sorted into 2D templates which were used to autopick 510,500 particles from all 2278 micrographs. Particles were ranked based on Z-score and 474,433 particles with a value of <0.8 were taken for subsequent rounds of 2D classification to remove false positives and suboptimal particles. The highest quality 108,354 particles based on the class image were refined to 5.5 Å using RELION 3D autorefine option without masking. The initial

model was the electron density map of lipid-bound TRPV5 (EMB-7965) filtered to 60 Å. A mask was made of this 5.5 Å map by filtering it to 15 Å at a threshold of 0.005, extending edge by five pixels and added a soft edge of 5 pixels. Additional rounds of 2D classification were performed on the original 474,433 particles and the best 97,702 particles were selected for 3D refinement using the 5.5 Å map as the initial model and masking the refinement with the mask described above. This produced s 4.9 Å map that was subjected to particle subtraction wherein a cylinder with a radius and height of 35 Å was isolated from the center of the TMD as described previously (*Hughes et al., 2018a*). These subtracted particles were 3D classified without applying angles and the best 46,384 particles were unsubtracted and subjected to several rounds of 3D refinement and 3D classification without applying angles. The final unsharpened map was composed of 21,802 particles and refined to 4.5 Å in RELION. The map was then sharpened using the software bfactor with an applied $b$ factor of $-330$ and final resolution was 4.2 Å as estimated by rmeasure (*Grigorieff, 2000*; *Sousa and Grigorieff, 2007*). Local resolution was calculating using RESMAP (*Kucukelbir et al., 2014*).

### ZINC17988990-bound TRPV5 structure determination

Motioncor2 was used to align each of the 1340 movies and adjust for beam induced motion (*Zheng et al., 2017*). CTF estimation was performed using GCTF on the summed movies (*Zhang, 2016*). Subsequent data processing was performed on the dose weighted micrographs in RELION (*Afonine et al., 2018*; *Kimanius et al., 2016*; *Scheres, 2016*) unless otherwise specified. Laplacian-of-Gaussian reference-free autopicking was used to pick 30,227 paricles from all 1340 micrographs. These particles were then 2D classified to create templates for autopicking. A total of 670,057 particles were autopicked from all 1340 micrographs. These particles were subjected to 2D classification and all particles within the best class with a defocus value >2.0 μm as estimated by GCTF were selected. These 2639 particles were refined to 15.7 Å without a mask using the electron density map of lipid-bound TRPV5 (EMB-7965) filtered to 60 Å as the initial model. A mask was created of this density map by applying a filter of 15 Å at a threshold of 0.004 and extending the edge and adding a soft edge of 7 pixels. The original 670,057 were 2D classified again to rescue more particles and the best 98,516 particles were refined with the mask described above and the filtered lipid-bound TRPV5 as the initial model. These particles were subjected to three rounds of CTF refinement, bayesian particle polishing and 3D refinement. The final unsharpened refinement was resolved to 4.06 Å and was sharped using the autosharpening feature in PHENIX (*Afonine et al., 2018*) to 3.78 Å. Local resolution was calculating using RESMAP (*Kucukelbir et al., 2014*).

## Model building and refinement

The previously published lipid-bound TRPV5 structure (PDB: 6DMR) was used as the initial model for the ZINC17988990-bound TRPV5 structure which was subsequently used as the initial model for the lower resolution ZINC9155420-bound TRPV5 structure. Each model was manually adjusted to fit their respective electron density maps in COOT (*Emsley and Cowtan, 2004*) and refined using phenix_realspace_refine (*Adams et al., 2010*) with imposed four-fold NCS. Compound restraint files for both compounds were created in PHENIX using the eLBOW (*Moriarty et al., 2009*) AM1 QM method of determining ligand geometry with the SMILES as the input. The SMILES can be found on the ZINC15 compound database (http://zinc15.docking.org/). Each compound was docked into the density map using COOT and the structure underwent several rounds of manual adjustment in COOT followed by refinement in PHENIX.

For model validation each model was randomized by 0.5 Å and refined against its respective half-map1. FSC curves of the randomized models vs each of its halfmaps and the final models vs their respective final map were created using EMAN2. Figures were made in PyMOL and Chimera (*Pettersen et al., 2004*). Pore diagrams were generating using HOLE (*Smart et al., 1996*). RMSD values of whole protein comparisons were calculated using PyMOL.

## Acknowledgements

We would like to acknowledge the Electron Microscopy Resource Laboratory at the University of Pennsylvania for use of cryo-EM screening equipment. We acknowledge Sabine Baxter at the Cell Center Service Facility at the University of Pennsylvania for assistance with hybridoma and cell

culture. cDNA clones for the various TRP channels were kindly provided by the following scientists: rbTRPV5: Dr. Rene Bindels, Radboud University, Nijmegen, Netherlands; hTRPV5: Dr. Jenny van der Wijst, Radboud University, Nijmegen, Netherlands; hTRPV6: Dr. Thomas McDonald, Albert Einstein College of Medicine; rTRPV4: Dr. Wolfgang Liedtke, Duke University; mTRPM3a2: Drs. Veit Flockerzi and Stephan Philipp, Universität des Saarlandes; rTRPM8 and rTRPV1: Dr. David Julius, UCSF. The drVSP clone was a gift from Dr. Bertil Hille, University of Washington. We highly appreciate the help of Dr. Paula Bartlett with the Flexstation 96-well plate reader. This research was, in part, supported by the National Cancer Institute's National Cryo-EM Facility at the Frederick National Laboratory for Cancer Research under contract HSSN261200800001E. This study was supported by grants from the National Institute of Health (R01GM103899 and R01GM129357 to VYM-B, R01GM093290 and R01NSNS055159 and R01GM131048 to TR). Computations were run on resources available through the Scientific Computing Facility at the Icahn School of Medicine at Mount Sinai and through the Extreme Science and Engineering Discovery Environment under MCB080077 (to MF), which is supported by National Science Foundation grant number ACI-1053575.

## Additional information

### Funding

| Funder | Grant reference number | Author |
|---|---|---|
| National Institute of General Medical Sciences | R01GM103899 | Vera Y Moiseenkova-Bell |
| National Institute of General Medical Sciences | R01GM129357 | Vera Y Moiseenkova-Bell |
| National Institute of General Medical Sciences | R01GM093290 | Tibor Rohacs |
| National Institute of General Medical Sciences | R01GM131048 | Tibor Rohacs |
| National Institute of Neurological Disorders and Stroke | R01NSNS055159 | Tibor Rohacs |
| National Science Foundation | ACI-1053575 | Marta Filizola |
| National Science Foundation | MCB080077 | Marta Filizola |

The funders had no role in study design, data collection and interpretation, or the decision to submit the work for publication.

### Author contributions

Taylor ET Hughes, Data curation, Formal analysis, Validation, Investigation, Visualization, Methodology, Writing—original draft, Writing—review and editing; John Smith Del Rosario, Abhijeet Kapoor, Data curation, Formal analysis, Validation, Investigation, Visualization, Methodology, Writing—review and editing; Aysenur Torun Yazici, Validation, Investigation, Methodology, Writing—review and editing; Yevgen Yudin, Formal analysis, Methodology; Edwin C Fluck III, Investigation, Visualization, Writing—review and editing; Marta Filizola, Resources, Supervision, Funding acquisition, Validation, Writing—review and editing; Tibor Rohacs, Resources, Supervision, Funding acquisition, Validation, Visualization, Writing—review and editing; Vera Y Moiseenkova-Bell, Conceptualization, Resources, Supervision, Funding acquisition, Validation, Writing—original draft, Writing—review and editing

### Author ORCIDs

Taylor ET Hughes ⓘ https://orcid.org/0000-0002-6064-1227
John Smith Del Rosario ⓘ https://orcid.org/0000-0003-4947-5835
Abhijeet Kapoor ⓘ https://orcid.org/0000-0002-3606-3463
Aysenur Torun Yazici ⓘ https://orcid.org/0000-0002-1715-0107
Edwin C Fluck III ⓘ http://orcid.org/0000-0002-7663-569X

Tibor Rohacs (iD) https://orcid.org/0000-0003-3580-2575
Vera Y Moiseenkova-Bell (iD) https://orcid.org/0000-0002-0589-4053

## Decision letter and Author response
Decision letter https://doi.org/10.7554/eLife.49572.041
Author response https://doi.org/10.7554/eLife.49572.042

# Additional files

## Supplementary files
• Transparent reporting form DOI: https://doi.org/10.7554/eLife.49572.031

## Data availability
The cryo-EM density maps and atomic coordinates of all structures presented in the text will be deposited into the Electron Microscopy Data Bank and Protein Data Bank under the following access codes: ZINC9155420-bound TRPV5 (PDB: 6PBF, EMB-20292); ZINC17988990-bound TRPV5 (PDB: 6PBE, EMB-20291).

The following datasets were generated:

| Author(s) | Year | Dataset title | Dataset URL | Database and Identifier |
|---|---|---|---|---|
| Hughes TET, Del Rosario JS, Kapoor A, Yazici AT, Yudin Y, Fluck EC, Filizola M, Rohacs T, Moiseenkova-Bell VY | 2019 | ZINC9155420-bound TRPV5 in nanodiscs | http://www.rcsb.org/structure/6PBF | Protein Data Bank, 6PBF |
| Hughes TET, Del Rosario JS, Kapoor A, Yazici AT, Yudin Y, Fluck EC, Filizola M, Rohacs T, Moiseenkova-Bell VY | 2019 | ZINC9155420-bound TRPV5 in nanodiscs | https://www.ebi.ac.uk/pdbe/entry/emdb/EMD-20292 | Electron Microscopy Data Bank, EMD-20292 |
| Hughes TET, Del Rosario JS, Kapoor A, Yazici AT, Yudin Y, Fluck EC, Filizola M, Rohacs T, Moiseenkova-Bell VY | 2019 | ZINC17988990-bound TRPV5 in nanodiscs | http://www.rcsb.org/structure/6PBE | Protein Data Bank, 6PBE |
| Hughes TET, Del Rosario JS, Kapoor A, Yazici AT, Yudin Y, Fluck EC, Filizola M, Rohacs T, Moiseenkova-Bell VY | 2019 | ZINC17988990-bound TRPV5 in nanodiscs | https://www.ebi.ac.uk/pdbe/entry/emdb/EMD-20291 | Electron Microscopy Data Bank, EMD-20291 |

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
