## [Decision Letter]

**Acceptance summary:**

The manuscript by Hughes and colleagues describes the identification of inhibitors of the TRPV5 ion channel using a structure-based virtual screen as a starting point to identify a set of molecules to screen in functional assays. This resulted in the identification of new potent inhibitors of TRPV5, including one that is specific for TRPV5 (when compared to the highly similar TRPV6, and several additional TRP channels). The authors also determine new structures of TRPV5 with these inhibitors (ZINC17988990) and localize a novel binding site in S1-S4. Functional studies show that mutations at this binding site dramatically reduce inhibition by ZINC17988990. Together, these studies identify a new druggable site of TRPV5 channel.

**Decision letter after peer review:**

[Editors’ note: this article was originally rejected after discussions between the reviewers, but the authors were invited to resubmit after an appeal against the decision.]

Thank you for submitting your work entitled "Structure-based discovery of novel TRPV5 inhibitors" for consideration by *eLife*. Your article has been reviewed by a Senior Editor, a Reviewing Editor, and three reviewers. The following individuals involved in review of your submission have agreed to reveal their identity: Rachelle Gaudet (Reviewer #2); Youxing Jiang (Reviewer #3).

Our decision has been reached after consultation between the reviewers. Based on these discussions and the individual reviews below, we regret to inform you that your work will not be considered further for publication in *eLife*.

Your study describes the discovery and structural characterization of a novel drug binding site on the TRPV5 channels. As you see will see below the reviewers have some concerns about the cryo-EM studies but are mainly concerned about the functional aspects of this study which does not rule out alternate mechanisms. The second major concern is that the study if anything demonstrates in silico screening does a poor job predicting the drug binding site although in your case the compound acts as an inhibitor in functional studies.

As a part of feedback to the authors, I am also appending some of the comments that were made during our discussion.

1) Where I do worry is with the M491A functional data. That is a region where we know in other TRPs we can find gating mutations. And they don't look at the effect of this mutation on other stimuli. Combined with the unconvincing density for the S4-S5 linker site, and the fact that as modeled, there are few contacts with protein, I'm not convinced that this is a real site.

2) I do think that they could/should focus the manuscript more on the positive results (especially the compound that binds to the S1-S4 domain). It would also be good to know what happens if they try to dock the econazole in the same pocket.

*Reviewer #1:*

This manuscript by Hughes et al. described the results of an in silico screen to find compounds in a library that would interact with the econozole binding site of TRPV5. They characterized one lead compound and one derivative of this compound for inhibitory effects on TRPV5 and TRPV6. The lead compound had μM affinity for both TRPV5 and TRPV6 and the derivative had subμM affinity for TRPV5 with no inhibition of TRPV6. CryoEM studies of TRPV5 bound to both compounds revealed binding sites for both compounds that did not overlap with the econozole binding site, demonstrating that the structure-based virtual screening was not ultimately useful.

The CryoEM studies, although not definitive, seem likely to have identified binding sites for the two compounds studied. This is a strength of the manuscript. Unfortunately, the functional studies do not directly address whether the identified binding sites indeed are functionally relevant nor the mechanism by which they inhibit activation. Mutations that shift the apparent affinity for a ligand need not be within the ligand binding site. An extreme example of this phenomenon is the voltage-sensing domain of voltage-gated ion channels; mutations throughout the sequence, even outside the membrane's electric field, shift the voltage dependence of activation. The authors need to show that mutations have no effects on the ligand-free gating of the channels and that they specifically affect regulation by ligands that act within these particular binding sites and not others. As it stands, the mutagenesis studies provide no mechanistic insight into the inhibitory effects of the compounds nor on the selectivity of the derivative compound.

*Reviewer #2:*

The manuscript by Hughes and colleagues describes the identification of inhibitors of the TRPV5 ion channel using a structure-based virtual screen as a starting point to identify a set of molecules to screen in functional assays. This resulted in the identification of new inhibitors of TRPV5, including one that is specific for TRPV5 (when compared to the highly similar TRPV6, and several additional TRP channels). The authors also determine new structures of TRPV5 and identify some densities that they attributed to the inhibitors, attributions that they support with IC50 measurements of mutant TRPV5 proteins.

In the maps provided, the densities assigned to ZINC17988990: The one in the S1-S4 seems quite strong, comparable to the protein density, and includes two regions that fit the aromatic groups well, with an appropriate distance and orientation so that the chosen pose fits very well. There is no comparable density in the other TRPV5 structures, including those from other groups. It is interesting to note this site partially overlaps with density observed in the econazole structure that was attributed to lipid in the NSMB publication.

I find the discovery of the ZINC17988990 compound and its binding site within the TRPV5 S1-S4 region exciting and compelling. Some of the other conclusions, however, are not as strongly supported by the data and analysis.

In particular:

The density for the ligands at the S4-S5 regions are rather weak (the ZINC17988990in particular), essentially gone in 5-σ maps, and weaker than several other unmodeled blobs of comparable or larger size. The modeled ligand poses are also somewhat strangely positioned; they contact very few protein atoms.

None of the modeled lipids are in the provided PDB: the modeled lipids should either not be discussed and illustrated in the figures, or they should be included in the model deposited in the PDB. This is particularly important because the authors also describe a SBVS in which they included these modeled lipids. On that note, the densities attributed to lipids seem much wider that I expect for a linear chain of methylene groups.

One of the densities assigned to lipid in this manuscript is very similar, and in the same position, to the density assigned to econazole in the NSMB publication. Can the authors comment on this somewhere in this paper?

The discussion on structural differences in subsection “Insights into the mechanism of TRPV5 inhibition by novel exogenous compounds” and the related figures (Figure 7 and Figure 8, and Supplementary figure 16 and Figure 5) are difficult to follow and evaluate, because the authors do not describe how they performed the superpositions of the different structures. The movements should instead be described as relative to otherwise. Also, RMSD values should include units (typically Å) and basis (Cα carbons, all atoms, which region of the protein, etc.?).

Discussion of the S2-S3 linker and the difference between TRPV5 and TRPV6: (and Figure 5—figure supplement 2) if there is no sequence difference between the two channels in that linker, then there should be some other sequence (or structure) difference somewhere else that explains why this linker is disordered in TRPV6 but ordered in TRPV5. Can the authors provide some hypothesis to that effect?

The ZINC91555420-bound structure is derived from such a small number of particles (~22,000) in comparison to the starting point (>500,000). Do the authors have any idea why that is the case? Are there potentially other structures represented in the other particles? Are there other classes identified?

Subsection “In silico compound screening”: the authors mention that no compound with similarities to econazole was identified in the screen. Did the library contain econazole itself? If the authors dock econazole, how does it score in comparison to the top compounds identified in the screen?

Subsection “Functional validation of compound hits”: the authors should introduce the "monovalent currents through TRPV5" – at least reference previous work that uses similar assays.

Along the same lines: Why didn't the authors look for activation (rather than only potentiation)? With the protocol the authors used, is the open probability of TRPV5 known (i.e. to what extent is that signal "potentiatable")?

It would be useful to also test the activation of other TRPs (beside TRPV5 and TRPV6) with ZINC9155420: the authors find that it is not selective between TRPV5 and TRPV6, and that raises the interesting question of whether this compound can also activate other TRPVs, and/or other TRPs (How broad or narrow is the selectivity?).

Could the authors test the combination of D406A and M491A and how this affects inhibition by ZINC17988990?

Related: it would be useful to have Supplementary figure 14 part of one of the main figures instead. And what about Y415F with ZINC9155420? It should not be assumed that Y415F would have less of an effect (if any) on inhibition by ZINC9155420 because its effect of this on inhibition by ZINC17988990 was less than that of D409A. That is not a given considering the chemical differences between the two inhibitor molecules.

*Reviewer #3:*

In this study, Hughes et al. performed a structure-based virtual screening (SBVS) of a large compound library focusing on a previously identified TRPV5 inhibitor (econazole) binding pocket. They identified two novel inhibitors of TRPV5, one of which is specific for TRPV5. They also define the binding sites of both inhibitors by determining the structures of TRPV5 in complex with each inhibitor using single particle cryo-EM. Since the exact position/orientation of the bound compounds as well as the interacting residues cannot be properly defined from the structures due to the resolution limit, the authors validated the binding sites by mutagenesis. Structural comparison between the inhibitor-bound closed TRPV5 and the PI(4,5)P2-bound open TRPV5 also provide structural insights into the allosteric inhibition of TRPV5 by these two inhibitors. This is a solid study with a large amount of structural and functional data. I have a few minor concerns.

1) If I understand correctly, the structure-based virtual screening (SBVS) was performed focusing on the previously identified econazole-binding pocket of TRPV5. However, the compounds identified from this screening actually do not bind to the predicted region. I am not sure if this can be considered as a successful example of an in silico screen as the actual binding site is different from the predicted/targeted binding site.

2) The binding sites for both compounds appear to be more accessible from the cytosolic side of the channel, particularly the TRPV5-specific ZINC17988990 that binds at the S1-S4 pocket. Should the authors use inside-out patch for recording in order to have a faster response of inhibition as well as a better/more accurate measurement of the IC50? Related to this, some compounds that have been tested that show no effect could be due to the compounds' inability to cross the membrane.

3) Figure 4B shows multiple nearby residues at the ZINC9155420-binding site but the authors only show the mutation of M491. I wonder if they have also tested other residues within this binding pocket.

---

## [Author Response]

Your study describes the discovery and structural characterization of of a novel drug binding site on the TRPV5 channels. As you see will see below the reviewers have some concerns about the cryo-EM studies but are mainly concerned about the functional aspects of this study which does not rule out alternate mechanisms. The second major concern is that the study if anything demonstrates in silico screening does a poor job predicting the drug binding site although in your case the compound acts as an inhibitor in functional studies.As a part of feedback to the authors, I am also appending some of the comments that were made during our discussion.1) Where I do worry is with the M491A functional data. That is a region where we know in other TRPs we can find gating mutations. And they don't look at the effect of this mutation on other stimuli. Combined with the unconvincing density for the S4-S5 linker site, and the fact that as modeled, there are few contacts with protein, I'm not convinced that this is a real site.

The M491A mutation had quite small effects, and as recommended in the second comment below, we have refocused the manuscript on the S1-S4 binding site for the specific derivative compound where our level of confidence is much higher. While we have still included this M491A data in our manuscript, we have made it clear for the derivative compound (ZINC17988990) that this site is unlikely to be involved in inhibition. Additionally, we were able to produce a M491A-D406A double mutant that supports our claim that the S4-S5 linker site does not play an appreciable role in ZINC17988990-mediated inhibition.

2) I do think that they could/should focus the manuscript more on the positive results (especially the compound that binds to the S1-S4 domain). It would also be good to know what happens if they try to dock the econazole in the same pocket.

We agree with this suggestion and have refocused the manuscript on the effect of the more specific compound (ZINC17988990) in the S1-S4 binding site. We also did attempt to fit econazole into the density seen for ZINC17988990, but the size of the molecule was too small to fit well, and the best fit we could identify yielded several clashes with protein density. This analysis was not included in the submitted manuscript.

Reviewer #1:This manuscript by Hughes et al., described the results of an in silico screen to find compounds in a library that would interact with the econozole binding site of TRPV5. They characterized one lead compound and one derivative of this compound for inhibitory effects on TRPV5 and TRPV6. The lead compound had μM affinity for both TRPV5 and TRPV6 and the derivative had subμM affinity for TRPV5 with no inhibition of TRPV6. CryoEM studies of TRPV5 bound to both compounds revealed binding sites for both compounds that did not overlap with the econozole binding site, demonstrating that the structure-based virtual screening was not ultimately useful.The CryoEM studies, although not definitive, seem likely to have identified binding sites for the two compounds studied. This is a strength of the manuscript. Unfortunately, the functional studies do not directly address whether the identified binding sites indeed are functionally relevant nor the mechanism by which they inhibit activation.

We respectfully disagree with the reviewer that the structure-based virtual screening was not ultimately useful, since the novel inhibitors would have not been discovered otherwise. We note that *in silico* screenings are not used to predict binding pockets, but rather to identify putative binders for predetermined pockets. As we reported in the manuscript, we would have prioritized for experimental validation the same series of compounds (out of ~12 million compounds) if we had assumed that the econazole binding site could be occupied by a lipid molecule instead of a small-molecule ligand. We have explained this discrepancy and offered a potential explanation for this in the revised manuscript. Specifically, we addressed this by re-screening the library in a grid that accounts for the presence of a lipid in the econazole binding pocket and concluded the following:

“Notably, a post hoc SBVS screen using (a) the ZINC9155420-bound TRPV5 cryo-EM atomic model with a lipid molecule in the econazole pocket and the other identified lipid in the vicinity, (b) the same ZINC15 compound library we used in the original screen, and (c) a screening grid centered on ZINC9155420, identified ZINC9155420 and ZINC17988990, as well as other derivatives of these two molecules, among the top-scored compounds. Therefore, while SBVS protocols utilize predefined binding sites, and we did not foresee lipid occupancy of the initially used econazole site, the retrospective identification of the same novel TRPV5 inhibitors characterized here if one used the correct binding pocket gives confidence in the approach used.”

Mutations that shift the apparent affinity for a ligand need not be within the ligand binding site. An extreme example of this phenomenon is the voltage-sensing domain of voltage-gated ion channels; mutations throughout the sequence, even outside the membrane's electric field, shift the voltage dependence of activation. The authors need to show that mutations have no effects on the ligand-free gating of the channels and that they specifically affect regulation by ligands that act within these particular binding sites and not others. As it stands, the mutagenesis studies provide no mechanistic insight into the inhibitory effects of the compounds nor on the selectivity of the derivative compound.

As we mentioned earlier, we refocused the revised manuscript on the S1-S4 binding site for the derivative compound, where the cryoEM resolution is best and the effects of the mutations are the most robust. Our data show that the D406A mutant robustly reduces inhibition by the derivative compound ZINC17988990 (Figure 4C). This is unlikely to happen via a non-specific effect, such increased open state stability of the channel, as this mutation did not reduce inhibition by the parent compound ZINC9155420 (Figure 4D). Nevertheless, we performed additional experiments to further address this issue, as described below:

The TRPV5 channel is a constitutively active channel, kept open by basal phosphatidylinositol 4,5-bisphosphate (PIP2) levels in the plasma membrane. We performed experiments with the voltage-inducible phosphoinositide phosphatases drVSP, and tested if the D406A and Y415F mutations have any effect on inhibition by depletion of PIP2. We found that inhibition of the D406A mutant was less than the inhibition of the wild type channel; the extent of this difference however was much less than the robust left shift of the ZINC17988990 dose response. The altered inhibition by PIP2 depletion is likely explained by the drug’s interference with PIP2 activation of the channel, which was discussed in the original manuscript as well as in this revised version. The inhibition of the Y415F mutant did not show a statistically significant difference from inhibition of the wild type TRPV5. Overall our original and new data strongly support that the D406 and Y415 residues are part of the functionally important binding site for ZINC17988990. The new data can be seen in Figure 5—figure supplement 1.

Reviewer #2:In the maps provided, the densities assigned to ZINC17988990: The one in the S1-S4 seems quite strong, comparable to the protein density, and includes two regions that fit the aromatic groups well, with an appropriate distance and orientation so that the chosen pose fits very well. There is no comparable density in the other TRPV5 structures, including those from other groups. It is interesting to note this site partially overlaps with density observed in the econazole structure that was attributed to lipid in the NSMB publication.I find the discovery of the ZINC17988990 compound and its binding site within the TRPV5 S1-S4 region exciting and compelling. Some of the other conclusions, however, are not as strongly supported by the data and analysis.In particular: The density for the ligands at the S4-S5 regions are rather weak (the ZINC17988990in particular), essentially gone in 5-σ maps, and weaker than several other unmodeled blobs of comparable or larger size. The modeled ligand poses are also somewhat strangely positioned; they contact very few protein atoms.

We agree with the reviewer that the ZINC17988990 compound and its binding site within the TRPV5 S1-S4 region was the strongest part of the manuscript, and have refocused the revised manuscript on this key finding.

Given the lower resolution of the densities in the S4-S5 binding site, and the smaller effects of the mutations in this site, our confidence level of this binding site is lower. To be absolutely conclusive with this binding site would require extensive additional work, including probably new structures, which we believe is beyond the scope of a revised manuscript. We also feel that this is unnecessary, as the focus of the manuscript will be the higher affinity, more specific compound the ZINC17988990, for which we have strong data indicating that most, if not all of its effects are exerted via binding the S1-S4 binding site. As such, we have provided a discussion on the limitation of the data with the S4-S5 binding site.

None of the modeled lipids are in the provided PDB: the modeled lipids should either not be discussed and illustrated in the figures, or they should be included in the model deposited in the PDB. This is particularly important because the authors also describe a SBVS in which they included these modeled lipids. On that note, the densities attributed to lipids seem much wider that I expect for a linear chain of methylene groups.One of the densities assigned to lipid in this manuscript is very similar, and in the same position, to the density assigned to econazole in the NSMB publication. Can the authors comment on this somewhere in this paper?

At this resolution we cannot definitively identify which lipids are bound to the TRPV5 channel and as such we did not include the lipids in the deposited model. In order to address this concern, we have only included lipid densities rather than modeled lipids in the figures.

We have, also, mentioned lipid placement with regard to the bound econazole and other lipids seen in previous TRPV5 structures in the revised manuscript.

The discussion on structural differences in subsection “Insights into the mechanism of TRPV5 inhibition by novel exogenous compounds” and the related figures (Figure 7 and Figure 8, and Supplementary figure 16 and Figure 5) are difficult to follow and evaluate, because the authors do not describe how they performed the superpositions of the different structures. The movements should instead be described as relative to otherwise. Also, RMSD values should include units (typically Å) and basis (Cα carbons, all atoms, which region of the protein, etc.?). Discussion of the S2-S3 linker and the difference between TRPV5 and TRPV6: (and Figure 5—figure supplement 2) if there is no sequence difference between the two channels in that linker, then there should be some other sequence (or structure) difference somewhere else that explains why this linker is disordered in TRPV6 but ordered in TRPV5. Can the authors provide some hypothesis to that effect?

We have edited the manuscript to clearly indicate the method of superimposition in the text and have included appropriate units for our distance measurements.

We have included a brief discussion of the potential for differences in adjacent regions to yield this structural divergence in the S2-S3 linker.

The ZINC91555420-bound structure is derived from such a small number of particles (~22,000) in comparison to the starting point (>500,000). Do the authors have any idea why that is the case? Are there potentially other structures represented in the other particles? Are there other classes identified?

The density for bound ZINC91555420 was present from early in the reconstruction process so there is likely not a large mixture of bound and unbound populations. The particles were selected due to the higher resolution of the reconstructed map. It is possible that the lower resolution particles that were excluded were transition states, but the resolution was too poor to draw any conclusions. This explanation has been included in the revised manuscript.

Subsection “In silico compound screening”: the authors mention that no compound with similarities to econazole was identified in the screen. Did the library contain econazole itself? If the authors dock econazole, how does it score in comparison to the top compounds identified in the screen?

Econazole was not part of the ZINC subset of "in-stock" compounds we chose to use for virtual screening. However, a docking experiment conducted on econazole revealed a less favorable docking score (-5.9) compared to that of the novel inhibitor we discovered (ZINC9155420; docking score: -9.38). This information has been included in the revised manuscript.

Subsection “Functional validation of compound hits”: the authors should introduce the "monovalent currents through TRPV5" – at least reference previous work that uses similar assays.

We introduced references for assaying TRPV5 activity by measuring monovalent currents, and briefly explained the rationale for these measurements. We also confirmed that ZINC17988990 also inhibits Ca^2+^ flux though TRPV5 (but not other TRP channels) by measuring cytoplasmic Ca^2+^ levels in HEK293 cells transfected with TRPV5 and GCaMP6 (Figure 2—figure supplement 2).

Along the same lines: Why didn't the authors look for activation (rather than only potentiation)? With the protocol the authors used, is the open probability of TRPV5 known (i.e. to what extent is that signal "potentiatable")?

We assume the reviewer meant why we did not look for potentiation, rather than only inhibition. None of the compounds showed any potentiation, but the channels are close to fully active at this protocol, even though there are reports that certain stimuli can further increase activity, these effects are generally small, so the dynamic range of potentiation is well below that of inhibition.

It would be useful to also test the activation of other TRPs (beside TRPV5 and TRPV6) with ZINC9155420: the authors find that it is not selective between TRPV5 and TRPV6, and that raises the interesting question of whether this compound can also activate other TRPVs, and/or other TRPs (How broad or narrow is the selectivity?).

We feel that the characterization of ZINC9155420 is beyond the scope of this manuscript. Also, ZINC9155420 is far inferior to ZINC17988990 as a TRPV5 inhibitor, as it has lower potency, it does not discriminate between TRPV5 and TRPV6, and it is a lot less stable than ZINC17988990. The latter was not emphasized in the original manuscript, but we had to prepare fresh DMSO stock solutions before every experiment, as frozen stocks lost activity within a couple of days. Therefore, we concluded that this compound is difficult to work with reliably, and its further characterization will not be useful for the research community. We briefly mentioned the lack of stability in the methods section of the revised manuscript.

Instead we solidified our results with the more specific ZINC 17988990 and performed additional Ca^2+^ flux experiments to ensure that not only monovalent currents, but also Ca^2+^ flux is inhibited. These new data are shown in Figure 2—figure supplement 1. They confirm the specificity of ZINC 17988990 at 10 μM for TRPV5 over TRPV6, confirm its lack of effect on TRPV1, TRPM8 and TRPV4, and extend our data to show lack of effect on TRPM3.

Could the authors test the combination of D406A and M491A and how this affects inhibition by ZINC17988990?

We have performed this experiment and found that the inhibition of the D406A-M491A double mutant by ZINC17988990 was indistinguishable from the D406 mutant, indicating that the contribution of the site containing the M491 residue to TRPV5 inhibition by ZINC17988990 is negligible. The data are shown in Figure 4—figure supplement 4C.

Related: it would be useful to have Supplementary figure 14 part of one of the main figures instead. And what about Y415F with ZINC9155420? It should not be assumed that Y415F would have less of an effect (if any) on inhibition by ZINC9155420 because its effect of this on inhibition by ZINC17988990 was less than that of D409A. That is not a given considering the chemical differences between the two inhibitor molecules.

We have placed the data from Supplementary figure 14 in the main manuscript (Figure 4D). We have found that the inhibition of the Y415F mutant by drVSP activation was not significantly different from wild type TRPV5, therefore it is unlikely that it caused a right shift in the ZINC17988990 by non-specific effects (Figure 5—figure supplement 1).

Reviewer #3:In this study, Hughes et al. performed a structure-based virtual screening (SBVS) of a large compound library focusing on a previously identified TRPV5 inhibitor (econazole) binding pocket. They identified two novel inhibitors of TRPV5, one of which is specific for TRPV5. They also define the binding sites of both inhibitors by determining the structures of TRPV5 in complex with each inhibitor using single particle cryo-EM. Since the exact position/orientation of the bound compounds as well as the interacting residues cannot be properly defined from the structures due to the resolution limit, the authors validated the binding sites by mutagenesis. Structural comparison between the inhibitor-bound closed TRPV5 and the PI(4,5)P2-bound open TRPV5 also provide structural insights into the allosteric inhibition of TRPV5 by these two inhibitors. This is a solid study with a large amount of structural and functional data. I have a few minor concerns.1) If I understand correctly, the structure-based virtual screening (SBVS) was performed focusing on the previously identified econazole-binding pocket of TRPV5. However, the compounds identified from this screening actually do not bind to the predicted region. I am not sure if this can be considered as a successful example of an in silico screen as the actual binding site is different from the predicted/targeted binding site.2) The binding sites for both compounds appear to be more accessible from the cytosolic side of the channel, particularly the TRPV5-specific ZINC17988990 that binds at the S1-S4 pocket. Should the authors use inside-out patch for recording in order to have a faster response of inhibition as well as a better/more accurate measurement of the IC50?

This would be a problematic experiment for the following reasons. (1) channel activity shows a marked and full rundown in excised patches, due to the loss of PIP2 in the patch membrane. (2) We are set up for excised patch measurements in the *Xenopus* oocyte system, where the drugs have a different IC50 from that obtained in HEK cells (Figure 2—figure supplement 3). (3) From the practical standpoint for future in vivo experiments intracellular application is not practical, so an IC50 in that system would not be helpful for the vast majority of future studies.

Related to this, some compounds that have been tested that show no effect could be due to the compounds' inability to cross the membrane.

Even though all compounds were lipid soluble, this is a possibility, and we mentioned it in the revised version.

3) Figure 4B shows multiple nearby residues at the ZINC9155420-binding site but the authors only show the mutation of M491. I wonder if they have also tested other residues within this binding pocket.

We did not test other mutations, as we refocused the manuscript on the clear and robust data with the derivative compound ZINC17988990 and the S1-S4 binding site.